# Innovative Binocular Vision Testing for Phoria and Vergence Ranges Using Automatic Dual Rotational Risley Prisms

**DOI:** 10.3390/s25051604

**Published:** 2025-03-05

**Authors:** Hui-Rong Su, Yu-Jung Chen, Yun-Shao Hu, Chi-Hung Lee, Shang-Min Yeh, Shuan-Yu Huang

**Affiliations:** 1Ph.D. Program of Electrical and Communications Engineering, Feng Chia University, Taichung 407, Taiwan; leah860307@gmail.com (H.-R.S.); p06176@gm.jente.edu.tw (Y.-J.C.); 2Department of Ophthalmology, Chung Shan Medical University, Taichung 402, Taiwan; zkaae246@gmail.com; 3Department of Electrical Engineering, Feng Chia University, Taichung 407, Taiwan; chihlee@fcu.edu.tw; 4Department of Optometry, Central Taiwan University of Science and Technology, Taichung 402, Taiwan

**Keywords:** binocular visual function, automatic dual rotational Risley prisms (ADRRPs), vergence ranges, phoropter, positive fusional vergence (PFV), negative fusional vergence (NFV)

## Abstract

This study evaluated binocular visual function using automatic dual rotational Risley prisms (ADRRPs) to measure phoria and vergence ranges. Thirty-nine (mean age: 21.82 ± 1.10 years; age range: 20–24 years) healthy adults with normal binocular vision participated. Each underwent baseline refraction exams followed by phoria and vergence tests conducted using both a phoropter with Maddox rods and the ADRRPs. The results revealed a strong positive correlation between the two instruments for distance phoria (r = 0.959, *p* < 0.001) and near-phoria measurements (r = 0.968, *p* < 0.001). For vergence testing, positive fusional vergence (PFV) at distance showed a moderate-to-strong correlation for break points (r = 0.758, *p* < 0.001) and a moderate correlation for recovery points (r = 0.452, *p* < 0.001). Negative fusional vergence (NFV) at distance demonstrated a strong correlation for break points (r = 0.863, *p* < 0.001) and a moderate correlation for recovery points (r = 0.458, *p* < 0.01). Near-vergence testing showed moderate-to-strong correlations for break points (r = 0.777, *p* < 0.001) and recovery points (r = 0.623, *p* < 0.001). The inclusion of Bland–Altman analysis provides a more comprehensive evaluation of agreement between ADRRPs and the phoropter. While strong correlations were observed, systematic bias and LoA indicate that these methods are not perfectly interchangeable. The ADRRPs demonstrated potential for binocular vision assessment but require further validation for clinical application.

## 1. Introduction

Binocular vision testing plays a crucial role in diagnosing and managing conditions such as strabismus, convergence insufficiency, and other binocular dysfunctions, which can significantly impact patients’ quality of life [1]. These disorders are not only associated with visual discomfort and reduced efficiency but may also affect cognitive performance, work productivity, and academic achievement, particularly in children [2,3,4]. Therefore, accurate and reliable binocular vision assessment is essential for early diagnosis, effective treatment, and long-term management [5,6].

For decades, the phoropter has been widely used in clinical optometric practice to evaluate binocular vision functions, including refractive errors, phoria, and vergence ranges in adults [7,8]. While it remains an essential diagnostic tool, concerns have emerged regarding its test–retest reliability and measurement variability in both clinical and research settings [9,10,11]. The reliance on manual adjustments and subjective patient responses introduces inconsistencies, making it challenging to obtain reproducible measurements [12,13]. Several factors contribute to this variability. Fatigue, cognitive load, and patient comprehension of the testing process can influence responses, while examiner skill plays a crucial role in precise prism adjustments and alignment [14,15,16,17,18,19,20,21]. These factors increase the complexity of achieving accurate and repeatable assessments, particularly in cases involving complex binocular vision dysfunctions [22,23,24,25,26].

Beyond measurement variability, the ergonomic limitations of traditional phoropters also present challenges. Patients with atypical facial structures, limited neck mobility, or difficulty maintaining a stable posture may experience discomfort, which can impact test performance and measurement reliability [19,20]. Additionally, phoropters are less effective in evaluating binocular vision in populations with developmental disabilities, where alternative testing approaches may be necessary [27,28,29]. A significant limitation of traditional phoropter-based testing is the lack of objective measurements. Phoria and vergence range assessments rely on subjective patient feedback, which may not always reflect actual oculomotor responses, reducing measurement precision [17,18]. Studies indicate that phoropter-based vergence assessments exhibit only moderate reliability, posing challenges for consistency across sessions.

Traditional phoropters and trial frames further introduce variability due to reliance on subjective responses, affecting accuracy in assessing vergence ranges, fusional reserves, and phoria. Research highlights their moderate test–retest reliability, complicating reproducibility [6,11]. In both clinical and research settings, this inconsistency is particularly concerning, as accurate and repeatable measurements are essential for precise diagnosis, monitoring, and treatment planning [30].

The primary objective of this study was to evaluate the agreement between the automatic dual rotational Risley prisms (ADRRPs; manufactured by OrthoV Co., Ltd., Kaohsiung City, Taiwan) and the traditional phoropter in measuring phoria and vergence ranges, while also highlighting the convenience of ADRRPs and its applicability in binocular vision assessment. Our focus was on comparing the measurement outcomes from both instruments to assess their statistical relationship. Specifically, we analyzed the correlation between ADRRPs and phoropter measurements, while also conducting Bland–Altman analysis to determine their level of agreement and potential biases [31,32].

## 2. Materials and Methods

### 2.1. Subjects

This study was approved by the Second Research Ethics Review Committee of Chung Shan Medical University Hospital (Approval Number: CS2-22104) and was conducted in accordance with the Declaration of Helsinki. The study included healthy adults aged 20–40 years, recruited through public announcements at Chung Shan Medical University.

Prior to participation, all individuals provided written informed consent and disclosed personal information, including occupation and medical history.

All participants completed interviews and underwent baseline refraction examinations, which included assessments of medical history, ocular surgery, medication use, binocular refractive status, visual acuity, and visual function. The inclusion and exclusion criteria are detailed in Table 1.

### 2.2. Basic Refraction Examination

The basic refraction examination was conducted to confirm whether participants met the binocular refractive status and visual acuity inclusion criteria for the study. Standard instruments, including a phoropter and an LCD visual acuity chart (optotypes: Standard Snellen E letters; contrast: 100%), were used. The room illumination was approximately 450 lux, ensuring optimal visibility while minimizing glare. The examination followed a sequence of monocular subjective refraction, followed by binocular balancing using standard prism dissociation techniques, yielding monocular refractive power, visual acuity, and binocular visual balance and acuity. All participants had stereoacuity < 40 arc seconds.

The uncorrected spherical refractive error (Sphere) ranged from +0.00 to −11.00 D, while the cylindrical error (Cylinder) was less than −1.25 D. Visual acuity was measured at 6 m (distance) and 40 cm (near) using an LCD chart. To ensure best-corrected visual acuity (BCVA) and standardize refractive conditions, participants wore daily disposable spherical contact lenses, eliminating variability from different spectacle prescriptions. BCVA ≤ 0.1 logMAR was required for inclusion.

### 2.3. Binocular Visual Function Examination

The binocular visual function examination in this study included a phoria test using the Maddox rod method and a horizontal fusional vergence test. The primary instruments used were a phoropter, ADRRPs, and related testing tools, including the near Thorington Card, Maddox rod, and an LCD visual target system for fusion testing. The fusion testing included an “E” optotype for fusion assessment and a 20/30 Snellen letter for fusional vergence evaluation. Both distance (6 m) and near (40 cm) phoria and vergence tests were performed to ensure a comprehensive evaluation of binocular visual function.

### 2.4. Design of the ADRRPs

The ADRRPs (manufactured by OrthoV Co., Ltd., Kaohsiung City, Taiwan) are a head-mounted, automated optical system that utilizes Risley prisms to generate virtual images at varying positions based on prism rotation angles. The system consists of two sets of counter-rotating Risley prism pairs, one for each eye, with each pair composed of two 10Δ wedge glass components. Thus, a single set of Risley prisms can achieve a combined prism power ranging from 20Δ base-out (BO) to 20Δ base-in (BI) by summing the prism powers of the individual components. This design allows for precise adjustment of prism power, ranging from 40Δ base-out (BO) to 40Δ base-in (BI), enabling controlled manipulation of vergence demands during testing (Figure 1a–c). The prism power adjustment can reach a maximum rate of 2 prism diopters per second, but in this study, it was set to 1.5 prism diopters per second. The rotation angle accuracy of each prism is 0.02° per step, achieved through a gear and motor system. The prism power is smoothly and continuously adjusted, ensuring precise modifications during testing. Additionally, the prism power settings can be configured using the ADRRPs application on a mobile phone or tablet.

### 2.5. Examiner’s Role in ADRRPs Testing

The examiner is responsible for recording key measurement values during ADRRPs testing. These include the magnitude and direction of phoria (esophoria or exophoria) in prism diopters, as well as the break and recovery points for both positive fusional vergence (PFV) and negative fusional vergence (NFV). These measurements ensure consistency and allow for direct comparison with traditional phoropter-based testing methods.

### 2.6. Phoria Testing

ADRRPs, combined with a Maddox rod placed in front of the participant’s right eye, were used to observe a penlight target at a fixed distance. Through the Maddox rod, the right eye perceived the penlight as a vertical line of light, while the left eye saw it as a point light source. The Thorington card was used for near measurements, while the Maddox rod was used for both distance and near testing. In both distance and near testing, ADRRPs followed the Maddox rod method and was compared with phoropter measurements. Additionally, for near testing, ADRRPs measurements were also compared with the Thorington card results.

If the patient perceives the vertical line of light and the point light source coinciding when the prism power is set to zero, this indicates orthophoria (no deviation). If the vertical line of light appears to the left of the point light source, it indicates exophoria, while if the vertical line of light appears to the right of the point light source, it indicates esophoria (Figure 1d). The greater the separation between the vertical line of light and the point light source, the larger the degree of ocular deviation. In cases of exophoria or esophoria, ADRRPs automatically adjust the prism power until the vertical line of light and the point light source overlap. The prism power required to achieve this overlap represents the amount of ocular deviation. This process remains subjective, as it relies on the participant’s perception, with the examiner recording the final prism diopter value and direction.

### 2.7. Vergence Ranges Testing

Combining ADRRPs with visual targets on a fixed LCD display enables the assessment of convergence and divergence abilities. The participant wears the head-mounted ADRRPs device to observe the “E” target at 40 cm and 6 m.

Negative Fusional Vergence (NFV) Measurement: The prism power (BI) is gradually increased from 0Δ. When the participant perceives the “E” target splitting into two, the corresponding prism power is recorded as the NFV break point. Then, 4Δ BI is added, and the prism power is gradually reduced. When the participant sees the two “E” targets merging back into one, the corresponding prism power is recorded as the NFV recovery point.

Positive Fusional Vergence (PFV) Measurement: The prism power (BO) is gradually increased from 0Δ. When the participant perceives the “E” target splitting into two, the corresponding prism power is recorded as the PFV break point. Then, 4Δ BO is added, and the prism power is gradually reduced. When the participant sees the two “E” targets merging back into one, the corresponding prism power is recorded as the PFV recovery point.

However, we initially included blur point measurements but found that many participants had difficulty recognizing and accurately reporting them, resulting in inconsistent data. In contrast, break and recovery points were more reliably identified and reported. To enhance accuracy and consistency, we focused our analysis on these points.

### 2.8. Statistical Analysis

To ensure consistency and proficiency in each experimental step, all procedures in this study were performed by the same researcher. Data collected included participant demographics, refraction examination results, ocular refractive status, and binocular visual function assessments. Refractive data were converted into equivalent spherical power for analysis.

All research data were compiled in Microsoft Excel 2019 and subsequently imported into IBM SPSS Statistics 22 for statistical analysis. The agreement between binocular visual function measurements obtained using the phoropter and ADRRPs was assessed using Bland–Altman analysis alongside correlation testing [33]. The Shapiro–Wilk and D’Agostino tests indicated deviations from normal distributions, and Spearman correlation was applied along with other statistical analyses. Statistical significance was set at *p* < 0.05.

## 3. Results

The participants in this study were students from Chung Shan Medical University, with a total of 39 subjects, including 17 males and 22 females. Their ages ranged from 20 to 24 years, with an average age of 21.82 ± 1.10 years.

The average spherical power of the right eye (OD) was −4.35 ± 2.85 D, and the average cylindrical power was −0.68 ± 0.53 D. For the left eye (OS), the average spherical power was −4.27 ± 2.90 D, and the average cylindrical power was −0.65 ± 0.45 D. The average far-distance (6 m) ocular deviation was −0.18 ± 2.67 ∆ (exophoria), while the average near-distance (0.4 m) ocular deviation was −1.21 ± 6.89 ∆ (exophoria), as shown in Table 2.

After calculating refractive prescriptions using the equivalent spherical power formula, the average equivalent spherical power for the right eye (OD) was −4.66 ± 2.92 D, and for the left eye (OS) was −4.59 ± 2.96 D. When participants wore prescribed daily disposable contact lenses, the average best-corrected visual acuity (BCVA) was 0.00 ± 0.07 logMAR for the right eye (OD), 0.00 ± 0.06 logMAR for the left eye (OS), and −0.07 ± 0.04 logMAR for binocular vision (OU). Based on the paired *t*-test power analysis, the estimated required sample size is approximately 34 participants. This study included a total of 39 participants, which is generally considered sufficient for visual function research.

### 3.1. Comparison of Phoria Measurements Using ADRRPs and a Phoropter

For the distance phoria examination, Pearson correlation analysis revealed a strong positive correlation between the measurements obtained using ADRRPs and those obtained with a phoropter and Maddox rod [r(39) = 0.959, *p* < 0.001]. These results demonstrate a high correlation between these two methods for measuring distance phoria, as shown in Figure 2a. Both ADRRPs (*p* = 1.0) and the phoropter (*p* = 1.0) were normally distributed. Given these results, Pearson correlation was deemed appropriate for analyzing the relationship between these measurements.

To evaluate the agreement between measurement methods, we performed a Bland–Altman analysis to assess the interchangeability between ADRRPs and phoropter measurements by examining systematic bias and agreement limits, as shown in Figure 2b. The analysis revealed a mean difference of 0.31 prism diopters, indicating that ADRRP measurements tend to be slightly more exophoric than phoropter measurements. The limits of agreement (LoA) ranged from −1.23 to 1.84, with most differences falling within these bounds; however, some data points exceeded these limits. This finding highlights that ADRRP and phoropter measurements are not perfectly interchangeable, warranting further methodological investigation. In terms of correlation analysis, the Pearson correlation coefficient was 0.959, demonstrating a very strong positive correlation between ADRRPs and phoropter measurements (*p* = 8.13 × 10^−^^22^, highly significant), while the Spearman correlation coefficient was 0.878, also indicating a strong positive correlation (*p* = 2.03 × 10^−^^1^^3^, highly significant). These results confirm a strong association between the two measurement methods, though the presence of systematic bias suggests the need for further refinement in measurement methodology.

For the near-phoria examination, a strong positive correlation was found between ADRRPs and phoropter measurements [r(39) = 0.893, *p* < 0.001], confirming their strong association, as shown in Figure 3a. Additional statistical tests validated our findings. With a sample size of 39, the Shapiro–Wilk test indicated non-normal distributions for both ADRRPs (*p* = 7.74 × 10^−5^) and phoropter (*p* = 0.00074), justifying the use of Spearman correlation. Bland–Altman analysis showed a mean difference of −0.02, with limits of agreement from −3.40 to 3.36, as shown in Figure 3b. While most differences fell within these bounds, some exceeded them, suggesting systematic bias between the methods. Despite a strong Spearman correlation of 0.893 (*p* = 2.20 × 10^−^^14^), indicating a strong relationship, the presence of a mean difference between ADRRPs and phoropter measurements suggests these methods are not perfectly interchangeable. Further studies are needed to refine ADRRPs’ accuracy, reliability, and clinical applicability.

For the near-phoria examination, a Spearman correlation of 0.608 (*p* < 0.001) was found between ADRRPs and the near Thorington card combined with a Maddox rod, indicating a moderate relationship between the two methods, as shown in Figure 4a. To further validate our findings, additional statistical tests were conducted. With a sample size of 39, the Shapiro–Wilk test confirmed non-normal distributions for both ADRRPs (*p* = 7.74 × 10^−5^) and Thorington card measurements (*p* = 1.03 × 10^−6^), justifying the use of Spearman correlation. Bland–Altman analysis revealed a mean difference of 0.54, with limits of agreement ranging from −7.08 to 8.16, as shown in Figure 4b. Approximately 95% of the differences fell within these bounds, as expected, while some data points exceeded them. This does not necessarily indicate systematic bias, as bias is reflected in the mean difference between the methods. Despite a Spearman correlation coefficient of 0.608 (*p* = 4.03 × 10^−5^), indicating a moderate relationship, correlation alone does not confirm interchangeability. Further analysis is needed to assess the agreement and potential differences between the methods.

### 3.2. Horizontal Fusional Vergence Testing

Horizontal fusional vergence testing evaluates both positive fusional vergence (PFV) and negative fusional vergence (NFV). We compared the measurements obtained using ADRRPs and a phoropter at two distances: 6 m for distance testing and 40 cm for near testing. The comparison specifically examines the break points and recovery points recorded during the tests.

### 3.3. Positive Fusional Vergence

For distance testing, ADRRPs showed a moderate-to-high Pearson correlations with the phoropter for break points [r (39) = 0.758, *p* < 0.001] and a moderate correlation for recovery points [r (39) = 0.452, *p* < 0.001]. For near testing, moderate-to-high Pearson correlations were found for break points [r(39) = 0.817, *p* < 0.001] and recovery points [r(39) = 0.727, *p* < 0.001]. Normality tests indicated that the vergence-range data followed a normal distribution, justifying the use of Pearson correlation. Bland–Altman analysis showed moderate agreement between ADRRPs and the phoropter. Break points had mean differences of 0.3 (distance) and 0.6 (near), with LoA from −1.80 to 2.40 (distance) and −1.50 to 2.70 (near). Recovery points had slightly lower agreement, with mean differences of 0.4 (distance) and 0.5 (near), and LoA from −1.70 to 2.50 (distance) and −1.60 to 2.60 (near).

### 3.4. Negative Fusional Vergence

For distance testing, ADRRPs showed a moderate-to-high Pearson correlation for break points [r(39) = 0.863, *p* < 0.001] and a moderate correlation for recovery points [r(39) = 0.458, *p* < 0.01]. For near testing, moderate-to-high Pearson correlations were found for break points [r(39) = 0.777, *p* < 0.001] and recovery points [r(39) = 0.623, *p* < 0.001], as shown in Table 3. Normality tests confirmed that the vergence-range data followed a normal distribution, supporting the use of Pearson correlation. Bland–Altman analysis showed moderate agreement. Break points had mean differences of 0.5 (both distance and near), with LoA from −1.60 to 2.60. Recovery points had greater variability, with mean differences of 0.2 (distance) and 0.3 (near), and LoA from −1.90 to 2.30 (distance) and −1.80 to 2.40 (near). Overall, ADRRPs showed moderate agreement with the phoropter, but greater variability in recovery points suggests the need for further improvements.

## 4. Discussion

This study evaluated the feasibility of Automatic Dual Rotational Risley Prisms (ADRRPs) for assessing phoria and vergence, comparing its measurements with traditional methods such as phoropters and the Thorington card with a Maddox rod. The results indicate ADRRPs’ potential as an alternative tool for binocular vision assessment.

ADRRPs showed strong correlations with phoropter measurements for distance phoria (r = 0.959) and near-phoria (r = 0.968, *p* < 0.001). However, correlation alone does not confirm measurement agreement. Bland–Altman analysis revealed systematic differences, with ADRRPs measurements slightly higher than phoropter values, consistent with previous findings on heterophoria variability (Lam et al., 2005 [34]; Oh et al., 2020 [35]).

In vergence testing, ADRRPs demonstrated moderate-to-high correlations with phoropter measurements, with break and recovery points ranging from r = 0.452 to 0.863 for distance and r = 0.623 to 0.817 for near. While this supports ADRRPs’ clinical potential, Bland–Altman analysis showed systematic differences, indicating the methods are not fully interchangeable. The moderate-to-high correlations for near vergence recovery points suggest ADRRPs could be valuable for near-vision assessments, where precise vergence control is essential.

Despite its automated operation, which improves measurement convenience, ADRRPs still relies on subjective responses, limiting its objectivity (Casillas and Rosenfield, 2006 [36]; Oh et al., 2020 [35]). Further validation is needed to confirm its clinical applicability across different populations. Compared to traditional phoropters, ADRRPs enhances efficiency and standardization while maintaining compatibility with patient-based responses. Prior studies have shown that automated systems improve test consistency (Han et al., 2010 [37]), but unlike eye-tracking systems, ADRRPs does not provide real-time, objective vergence adaptation assessment (Mestre et al., 2018 [38]; Gantz and Caspi, 2020 [39]). Future improvements could integrate eye-tracking technology for enhanced precision.

For phoria measurements, the limits of agreement (LoA) between ADRRPs and phoropters were ±1.5 PD for distance and ±3.4 PD for near, while the Thorington card exhibited a wider LoA of ±7.5 PD for near-phoria. In vergence testing, the LoA was approximately ±2.1 PD at both distances, aligning with prior repeatability studies (Rainey et al., 1998; Anstice et al., 2021; Facchin and Maffioletti, 2021; Catherine McDaniel and Nick Fogt, 2010; Antona et al., 2008) [40,41,42,43,44]. While ADRRPs measurements showed strong agreement with traditional methods, its interchangeability remains limited due to methodological differences. Further research is necessary to refine its clinical application.

A key advantage of ADRRPs is its automated, head-mounted design, improving standardization and adaptability, particularly for patients with anatomical variations. However, its reliance on subjective responses and potential proximal convergence effects requires further investigation to enhance accuracy and usability.

### Limitations

Despite its benefits, ADRRPs still relies on subjective patient perception, unlike eye-tracking technologies that provide real-time, objective assessments (Mestre et al., 2018; Gantz and Caspi, 2020) [38,39]. As a head-mounted device, ADRRPs may also induce proximal convergence and accommodation, potentially affecting phoria and vergence measurements. Future research should evaluate this impact by comparing ADRRPs with objective gold-standard methods and explore strategies to minimize these effects.

ADRRPs show strong correlations with traditional tools, but further research is needed to validate their accuracy, reliability, and clinical applicability. Future advancements should integrate eye-tracking technology to enhance objectivity and real-time measurement capabilities.

## 5. Conclusions

In conclusion, the ADRRPs device offers a novel approach to binocular vision testing, particularly for evaluating phoria and vergence ranges. Its strong correlations with traditional methods suggest its potential as a complementary tool for clinical assessments. While its automated operation may improve testing efficiency, further research is necessary to validate its accuracy and reliability across diverse clinical settings and populations. Additionally, studies should explore its broader applications to determine its full clinical utility.

## 6. Patent

A Taiwan invention patent (I781072) has been registered for the automatic dual rotational Risley prisms system.

## Figures and Tables

**Figure 1 sensors-25-01604-f001:**
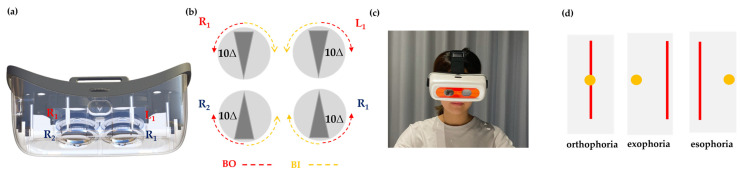
Schematic diagram of (**a**) the ADRRPs structure, (**b**) two pairs of Risley prisms and their rotation modes, (**c**) a participant wearing ADRRPs during phoria and vergence range testing, and (**d**) the observation of the relative position between the penlight spot and the light beam when a Maddox rod is placed in front of the eye.

**Figure 2 sensors-25-01604-f002:**
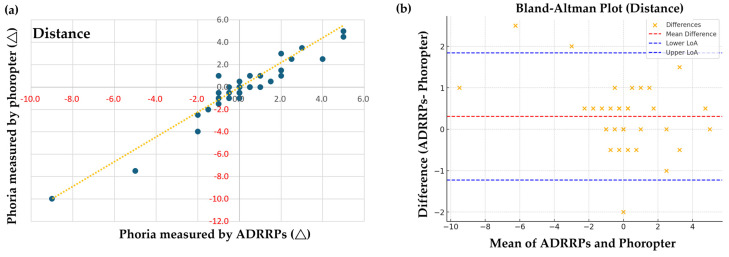
(**a**) Pearson correlation analysis of distance phoria measured with ADRRPs and a phoropter with a Maddox rod. Phoria is measured in prism diopters (∆), with “+” indicating esophoria and “−” indicating exophoria. (**b**) Bland–Altman analysis of distance phoria, assessing the interchangeability of ADRRPs and phoropter measurements.

**Figure 3 sensors-25-01604-f003:**
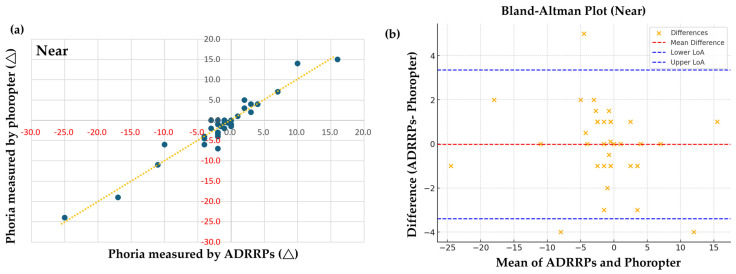
(**a**) Pearson correlation analysis of near-phoria measured with ADRRPs and a phoropter with a Maddox rod. Phoria is measured in prism diopters (∆), with “+” indicating esophoria and “−” indicating exophoria. (**b**) Bland–Altman analysis of near-phoria, assessing the interchangeability of ADRRPs and phoropter measurements.

**Figure 4 sensors-25-01604-f004:**
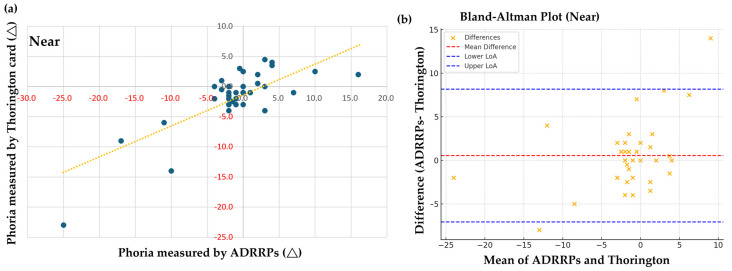
(**a**) Pearson correlation analysis of near-phoria measured with ADRRPs and a Thorington card with a Maddox rod. Phoria is measured in prism diopters (∆), with “+” indicating esophoria and “−” indicating exophoria. (**b**) Bland–Altman analysis of near-phoria, evaluating the interchangeability of ADRRPs and the Thorington card with a Maddox rod.

**Table 1 sensors-25-01604-t001:** Inclusion and Exclusion Criteria for Participants.

Inclusion Criteria (All of the Following Criteria Must Be Met)	Exclusion Criteria (Participants Will Be Excluded if Any of the Following Conditions Apply)
Age: 20–40 years Spherical refractive error between +0.00 and −11.00 Astigmatic refractive error ≤ −1.25 Best corrected visual acuity (BCVA) in one eye ≤0.1 logMAR Normal binocular visual function	Presence of ocular diseases Presence of systemic diseases History of ocular surgery Use of ocular-related medications History of strabismus or amblyopia

**Table 2 sensors-25-01604-t002:** Baseline of the Participants.

Participants (N = 39)	
Characteristics	Mean ± SD
Gender, (F/M)	22/17
Age (y)	21.82 ± 1.10
D Phoria (∆)	−0.18 ± 2.67
N Phoria (∆)	−1.21 ± 6.89
Spherical Equivalent (D)	
OD	−4.66 ± 2.92
OS	−4.59 ± 2.96
BCVA (logMAR)	
OD	0.00 ± 0.07
OS	0.00 ± 0.06
OU	−0.07 ± 0.04

F—female; M—male; D—distance; N—near; OD—right eye; OS—left eye; OU—binocular vision; BCVA—best corrected visual acuity; SD—standard deviation.

**Table 3 sensors-25-01604-t003:** Pearson Correlation Analysis of Positive Fusional Vergence (PFV) and Negative Fusional Vergence (NFV).

Measurements				ADRRPs	
			Break Point		Recovery Point
Phoropter	PFV	Distance	0.758 ***		0.452 ***
		Near	0.817 ***		0.727 ***
	NFV	Distance	0.863 ***		0.458 ***
		Near	0.777 ***		0.623 ***

*** *p* < 0.001.

## Data Availability

The datasets generated during and/or analyzed during the current study are available from the corresponding author on reasonable request.

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
