# Peer review of "Innovative Binocular Vision Testing for Phoria and Vergence Ranges Using Automatic Dual Rotational Risley Prisms"

_sensors, 2025, doi:10.3390/s25051604_

Round 1
Reviewer 1 Report
Comments and Suggestions for Authors
This study compared the phoria and vergence ranges measured with Maddox Rod (distance and near) + phoropter, and Maddox Rod + Thorington card for near clinical tests, to a novel automatic dual rotational Risley prism device that is head mounted, in 39 participants between the ages of 20-40 (mean age: 22). The experiment examines if the methods interchangeable. However, the authors are impressed that they examined reliability, validity, accuracy, precision, convenience though none of these were directly addressed in the study. Below I have provided several references that can facilitate the authors in acknowledging how to address reliability ,validity, accuracy, precision, and convenience of testing. I have also provided references for the authors to apply appropriate analyses to their data to address interchangeability.
The manuscript cannot be published in its current form and requires extensive rewriting and editing. Major and minor comments stated below. The experimental question needs to be rewritten, some methods detailed more in depth, some minor corrections and data re-analysis, and a complete rewriting of the discussion.
In terms of methods, the authors do not provide enough information about the device and a photograph would have been useful. They also do not provide enough detail on the the assessments that were carried out to allow replication of the experiment. For example, who adjusts the prisms in the automatic dual rotation device? The user? The tester? The discussion should compare findings of the present study to past experiments comparing phoropter based results to trial-frame based results, and comparing clinical assessments to truly objective eye-tracking measurements. The discussion should also include limitations of the study and (if I understood correctly how the device works) address the potential proximal vergence and accommodation it may induce.
Major comments:
Abstract
Conclusion not supported by findings. The authors report a correlation. This suggests a relationship or association between variables but does not exhibit “accuracy, reliability, and effectiveness”. These must be examined differently. Reliability can be assessed using inter-rater reproducibility and inter-session reproducibility. Accuracy can be assessed by repeating the measurement on the same participant several times and examining the standard deviation and by comparing results with a known gold standard. In the case of phoria and fusional reserves, I would argue that an eye tracker would be a true objective gold standard. More information on these terms and methods to assess them can be found in this paper:
McAlinden, Colm MB BCh, BSc (Hons), MSc, PhD; Khadka, Jyoti PhD; Pesudovs, Konrad PhD. Precision (repeatability and reproducibility) studies and sample-size calculation. Journal of Cataract & Refractive Surgery 41(12):p 2598-2604, December 2015. | DOI: 10.1016/j.jcrs.2015.06.029
In any case, based on the analysis undertaken by the authors, they can only attest that the two measurement methods are associated with each other significantly.
Introduction
Page 71 the study evaluated the interchangeability of the methods. It did not examine the “advancement in performance” and did not examine reliability
Methods
Methods described in a research paper must allow replication. Missing description of which assessments constitute “binocular visual balance and acuity”? What were the normative expected values? Which assessments were performed, at what distance, and which visual targets? Were they performed with habitual correction?
What distance was visual acuity measured and with which correction (described in Table 1 and not in text)?
Just to clarify: the ADRRP Phoria testing and vergence is still subjective and requires patient perception and patient responses? The only difference is that the adjustment of the prisms is performed automatically without the examiner intervention?
Does the examiner also record the amount and direction of phoria? And the amount of prisms for break and recovery?
Statistics
Missing sample size calculation to indicate the power of the findings
Missing examination of normality distribution of the findings
Pearson for normal distributions, Spearman for non-normal distributions
Assuming also descriptive statistics were employed to describe the study sample. If not, they should be
Missing Bland-Altman (parametric or non-parametric) analysis to examine the interchangeability between the two measurement methods, and if there is a bias
Line 253 instead of explored the use: examined the feasibility of using Automatic Dual Rotational Risley Prisms to evaluate phoria and vergence ranges.
Vergence ranges is the clinical term
Lines 255-256 but you also compared to the Maddox and Thorington test in free space so you did not just compare it to the traditional phoropter. Perhaps state that it was compared with traditional testing methods.
Your results demonstrated an association between two methods of measuring the same thing. If you read Bland & Altman (1986)’s paper, you will see that this is to be expected.
Bland, J. M., & Altman, D. G. (1986). Statistical methods for assessing agreement between two methods of clinical measurement. Lancet (London, England), 1(8476), 307–310.
However, your results DID NOT suggest that the ADRRPs device has significant potential in improving the precision, reliability, and convenience of binocular vision as- sessments.
Precision- should have been compared to an eye tracker which is “truth”
Reliability- inter-examiner or inter-session reproducibility, and even calculating the standard deviation of repeated measurements within the same participant would have provided a measure of reliability
Convenience- could have been examined by asking the participants and examiners which method they found more convenient
The only conclusion based on the correlations presented herein, is that the two methods are associated with each other. The authors have the data to run Bland-Altman analysis to determine interchangeability between the measurement methods.
Line 259: strong positive correlations do not validate accuracy unless the new device is strongly correlated with the gold standard. None of the existing tests can be said to be the gold standard.
Line 263: the ADRRP is not operator dependent? Who controls the application?
It is also worth mentioning if the ADRRP is patient- dependent because the patient has to provide input regarding the end point of the test.
Line 270: the correlation does not support effectiveness of a method
Line 277- you can only claim that the patient has improved comfort if you actually asked the patient about the comfort with the two methods and compared, Otherwise it’s just an unfounded conclusion.
How do the findings of this study compare with other studies comparing between phoropter and free space measurements of binocular visual functions?
For example:
Lam, A. K., Lam, A., Charm, J., & Wong, K. M. (2005). Comparison of near heterophoria tests under varying conditions on an adult sample. Ophthalmic and Physiological Optics, 25(2), 162-167.
Casillas Casillas E, Rosenfield M. Comparison of subjective heterophoria testing with a phoropter and trial frame. Optom Vis Sci. 2006 Apr;83(4):237-41. doi: 10.1097/01.opx.0000214316.50270.24. PMID: 16614580.
Tsotetsi, A. L., & Mathebula, S. D. (2021). Comparison of phoropter and trial frame-based von Graefe heterophoria measures in non-presbyopic participants. African Vision and Eye Health, 80(1), 645.
Oh, K. K., Cho, H. G., Moon, B. Y., Kim, S. Y., & Yu, D. S. (2020). Change in lateral phoria under a phoropter and trial frame in phoria tests. J Korean Ophthalmic Opt Soc, 25(4), 395-403.
How do the findings of this study compare with other studies comparing between phoropter and free space measurements and eye tracking measurement of binocular visual functions?
Han, S. J., Guo, Y., Granger‐Donetti, B., Vicci, V. R., & Alvarez, T. L. (2010). Quantification of heterophoria and phoria adaptation using an automated objective system compared to clinical methods. Ophthalmic and Physiological Optics, 30(1), 95-107.
Mestre, C., Otero, C., Díaz-Doutón, F., Gautier, J., & Pujol, J. (2018). An automated and objective cover test to measure heterophoria. PloS one, 13(11), e0206674.
Gantz, L., & Caspi, A. (2020). Synchronization of a removable optical element with an eye tracker: test case for heterophoria measurement. Translational vision science & technology, 9(7), 40-40.
The manuscript is missing study limitations.
There are disadvantages in the ADRRP that the authors fail to mention. Is the ADRRP worn on the head like VR goggles ? If so, this induces proximal convergence and proximal accommodation which can affect the phoria and vergence measurements.
The manuscript can be further strengthened with an image of the ADRRP and how it is worn.
Minor comments:
Abstract
Using automatic dual rotating without the word “an”
State the mean age and age range of the participants in parenthesis in the abstract
Introduction
Line 34- need a reference for strabismus and binocular visual dysfunctions impacting quality of life
Line 39, and again paragraph starting with Line 53- if the authors are referring to children, this is not an accurate statement. Pediatric testing is typically performed with trial frames and not phoropter
Paragraph 60: this is true also for trial frames… this is the downside of most optometric tests
Methods
Line 80 The study approval should be the first sentence in the subjects
Line 79- were they consented before they provided information about their occupation and basic eligibility testing?
Also, line 79 “Participants provided and overview of their ocular history and underwent a basic refraction examination for eligibility screening” is redundant with lines 84-85: “All participants completed interviews and refraction exams, including assessments of medical history, ocular surgery, medications, binocular refractive status, visual acuity, and visual function.”
Maybe instead of “Basic Refraction Examination” replace with “Baseline Refraction Examination”
Line 100: phoria test using the Maddox Rod method.
Was this at distance or near? Typically the Thorington card is used for near, but distance versions do exist. What was used here? From the results it appears that for distance ONLY a Maddox Rod was used, and for near the Maddox Rod was used in conjunction with a Thorington card OUTSIDE of the phoropter and was measured with JUST the Maddox Rod in the phoropter. However, I am gathering this assumption mainly from Figure 2 and 3, not from the methodology where it should be clearly stated.
For the ADRRP method, does the method replicate the two clinical tests exactly? Or does the near ADRRP method include just the Maddox Rod?
Line 119: remove the word “seamless”- which is typically used for publicity and is not evidence based
Line 131 this indicates (to be consistent with previous sentence)
Line 132: this indicates (for consistency)
Also curious as to why the authors did not include measurement of the blur point on the vergence ranges?
Results (with an S)
Line 197 indicates instead of confirms as the authors did not state any hypothesis to be confirmed
Line 219-220 and 221-222: positive significant correlation- then you can delete “with statistical significance”
Discussion
Table 1:
Instead of “Ocular health: Participants must have..” just write: free of ocular pathology, though it also appears in the exclusion and can be removed from the inclusion criteria altogether
Table 2: Demographic information of the study participants. Also provide a key describing what the letters represent. F-female, M- male, D-distance, N-near, OD- right eye, OS- left eye, BCVA- best corrected visual acuity, SD- standard deviation
Combine Tables 3 and 4 into one table
Comments on the Quality of English Language
The English is overall written well. Specific minor comments were specified.
Author Response
Subject: Response to Reviewer – Manuscript No. sensors-3454806
Dear Reviewer,
Thank you for your valuable time and insightful comments on our manuscript titled "Innovative Binocular Vision Testing for Phoria and Fusional Ability Using Automatic-Dual-Rotational-Risley-Prisms." We appreciate your constructive feedback, which has helped us refine and strengthen our work.
Below, we have carefully addressed each of your comments and incorporated the necessary revisions into the manuscript. For ease of reference, we provide a point-by-point response, highlighting the changes made.
Question 1: This study compared the phoria and vergence ranges measured with Maddox Rod (distance and near) + phoropter, and Maddox Rod + Thorington card for near clinical tests, to a novel automatic dual rotational Risley prism device that is head mounted, in 39 participants between the ages of 20-40 (mean age: 22). The experiment examines if the methods are interchangeable. However, the authors are impressed that they examined reliability, validity, accuracy, precision, and convenience though none of these were directly addressed in the study. Below I have provided several references that can facilitate the authors in acknowledging how to address reliability, validity, accuracy, precision, and convenience of testing. I have also provided references for the authors to apply appropriate analyses to their data to address interchangeability.
The manuscript cannot be published in its current form and requires extensive rewriting and editing. Major and minor comments stated below. The experimental question needs to be rewritten, some methods detailed more in depth, some minor corrections and data re-analysis, and a complete rewriting of the discussion.
In terms of methods, the authors do not provide enough information about the device and a photograph would have been useful. They also do not provide enough detail on the assessments that were carried out to allow replication of the experiment. For example, who adjusts the prisms in the automatic dual rotation device? The user? The tester? The discussion should compare findings of the present study to past experiments comparing phoropter based results to trial-frame based results and comparing clinical assessments to truly objective eye-tracking measurements. The discussion should also include limitations of the study and (if I understood correctly how the device works) address the potential proximal vergence and accommodation it may induce.
Response: Thank you for your valuable feedback. We have thoroughly revised the entire manuscript to enhance its clarity and scientific rigor. In the following sections, we have carefully addressed your comments on this specific part, providing detailed explanations and corresponding modifications to ensure our revisions are aligned with your suggestions. We appreciate your insightful input, which has helped us improve the quality of our work.
Major comments:
Question 2: Abstract-Conclusion not supported by findings. The authors report a correlation. This suggests a relationship or association between variables but does not exhibit “accuracy, reliability, and effectiveness”. These must be examined differently. Reliability can be assessed using inter-rater reproducibility and inter-session reproducibility. Accuracy can be assessed by repeating the measurement on the same participant several times and examining the standard deviation and by comparing results with a known gold standard. In the case of phoria and fusional reserves, I would argue that an eye tracker would be a true objective gold standard. More information on these terms and methods to assess them can be found in this paper:
McAlinden, Colm MB BCh, BSc (Hons), MSc, PhD; Khadka, Jyoti PhD; Pesudovs, Konrad PhD. Precision (repeatability and reproducibility) studies and sample-size calculation. Journal of Cataract & Refractive Surgery 41(12): p2598-2604, December 2015. | DOI: 10.1016/j.jcrs.2015.06.029
In any case, based on the analysis undertaken by the authors, they can only attest that the two measurement methods are associated with each other significantly.
Response: We appreciate the reviewer’s detailed feedback and agree that our study primarily demonstrates a significant association between the two measurement methods rather than directly establishing accuracy, reliability, or effectiveness. Accordingly, we have revised the abstract to ensure our conclusions align with the findings.
Regarding reliability, we acknowledge that inter-rater and inter-session reproducibility are standard assessment approaches. While our study showed strong correlations between ADRRPs and the phoropter, a more comprehensive evaluation, such as test-retest reproducibility, is needed for definitive conclusions.
In response, we have revised the abstract to remove definitive claims about accuracy, reliability, and effectiveness, focusing instead on the observed statistical associations between ADRRPs and the phoropter. We appreciate the reviewer’s insightful critique, which has helped improve the clarity and rigor of our manuscript.
“The ADRRPs showed a strong correlation with the phoropter for phoria measurements and a moderate-to-high correlation for vergence testing, indicating its potential for binocular vision assessment.” in lines 26-28.
Question 3: Introduction-Line 71 the study evaluated the interchangeability of the methods. It did not examine the “advancement in performance” and did not examine reliability.
Response: We appreciate the reviewer’s comments and would like to clarify the study’s scope. Our primary goal was to assess the interchangeability of automatic dual rotational Risley prisms (ADRRPs) and the phoropter in measuring phoria and vergence abilities. We focused on comparing measurement consistency rather than evaluating performance advancements or reliability improvements. Our findings showed a high correlation between ADRRPs and the phoropter, supporting their interchangeability but not suggesting ADRRPs outperform the phoropter. We are willing to revise the Introduction to explicitly clarify this focus if needed.
“The primary objective of this study was to evaluate the agreement between the automatic dual rotational Risley prisms (ADRRPs) and the traditional phoropter in measuring phoria and vergence ranges. Our focus was on comparing the measurement outcomes from both instruments to assess their statistical relationship. Specifically, we analyzed the correlation between ADRRPs and phoropter measurements, while also conducting Bland-Altman analysis to determine their level of agreement and potential biases [31,32].”
[30, 31].” in lines 76-81.
Question 4: Methods-Methods described in a research paper must allow replication. Missing description of which assessments constitute “binocular visual balance and acuity”? What were the normative expected values? Which assessments were performed, at what distance, and which visual targets? Were they performed with habitual correction?
What distance was visual acuity measured and with which correction (described in Table 1 and not in text)?
Just to clarify: the ADRRP Phoria testing and vergence is still subjective and requires patient perception and patient responses? The only difference is that the adjustment of the prisms is performed automatically without the examiner intervention?
Does the examiner also record the amount and direction of phoria? And the amount of prisms for break and recovery?
Responses: Thank you for the reviewer's comments. We have provided a brief explanation here, and the detailed revisions have been fully incorporated into the Methods section.
- Binocular Visual Balance and Acuity Assessments: (a)Binocular balance: Monocular subjective refraction followed by prism dissociation. (b)Visual acuity: Measured at 6m (distance) and 40cm (near) using an LCD chart. (c)Normative values: ≤0.1 logMAR for study inclusion.
- Assessments, Testing Distances, and Visual Targets: (a)Phoria: Measured at 6m (Maddox rod & penlight) and 40cm (Thorington card). (b)Vergence: PFV & NFV tested at 6m & 40cm with Risley prisms and ADRRPs. (c)Visual Targets: “E” optotype for fusion; 40/60 Snellen letter for vergence.
- Habitual Correction: All tests were performed with best-corrected vision using daily contact lenses.
- Visual Acuity Measurement: 6m (distance) and 40cm (near) using an LCD chart, with soft contact lenses for consistency.
- ADRRPs Phoria and Vergence Testing: Still subjective patients align images, but prism adjustments are automated instead of manual.
- Examiner's Role (a)Recorded phoria magnitude & direction (prism diopters). (b)Noted break & recovery points for PFV & NFV.
Question 5: Statistics-Missing sample size calculation to indicate the power of the findings. Missing examination of normality distribution of the findings. Pearson for normal distributions, Spearman for non-normal distributions. Assuming also descriptive statistics were employed to describe the study sample. If not, they should be. Missing Bland-Altman (parametric or non-parametric) analysis to examine the interchangeability between the two measurement methods, and if there is a bias.
Response: We have incorporated a revised statistical approach based on the reviewer's feedback and have detailed the methodology in the Methods section. The statistical results have been thoroughly analyzed and are presented in both the Results and Discussion sections to ensure clarity and comprehensive interpretation of the findings.
Statistical Analysis
“To ensure consistency and proficiency in each experimental step, all procedures in this study were performed by the same researcher. Data collected included participant demographics, refraction examination results, ocular refractive status, and binocular visual function assessments. Refractive data were converted into equivalent spherical power for analysis.
All research data were compiled in Microsoft Excel 2019 and subsequently imported into IBM SPSS Statistics 22 for statistical analysis. The agreement between binocular visual function measurements obtained using the phoropter and ADRRPs was assessed using Bland-Altman analysis alongside correlation testing. As normality tests (Shapiro-Wilk and D’Agostino tests) indicated non-normal distributions, Spearman correlation was applied along with other statistical analyses. Statistical significance was set at p < 0.05.” in lines 177-187.
Question 6: Line 253 instead of explored the use: examined the feasibility of using Automatic Dual Rotational Risley Prisms to evaluate phoria and vergence ranges. Vergence ranges is the clinical term.
Response: Thank you for the reviewer's comments. We have made revisions based on the suggestions.
“This study examined the feasibility of using Automatic Dual Rotational Risley Prisms (ADRRPs) for assessing phoria and vergence.” in lines 306-307.
Question 7: Lines 255-256 but you also compared to the Maddox and Thorington test in free space so you did not just compare it to the traditional phoropter. Perhaps state that it was compared with traditional testing methods.
Your results demonstrated an association between two methods of measuring the same thing. If you read Bland & Altman (1986)’s paper, you will see that this is to be expected.
Bland, J. M., & Altman, D. G. (1986). Statistical methods for assessing agreement between two methods of clinical measurement. Lancet (London, England), 1(8476), 307–310.
However, your results DID NOT suggest that the ADRRPs device has significant potential in improving the precision, reliability, and convenience of binocular vision assessments.
Precision- should have been compared to an eye tracker which is “truth”
Reliability- inter-examiner or inter-session reproducibility, and even calculating the standard deviation of repeated measurements within the same participant would have provided a measure of reliability
Convenience- could have been examined by asking the participants and examiners which method they found more convenient
The only conclusion based on the correlations presented herein, is that the two methods are associated with each other. The authors have the data to run Bland-Altman analysis to determine interchangeability between the measurement methods.
Response: Thank you for the reviewer's insightful comments. Below are our responses and corresponding revisions.
- Clarification on Comparison with Traditional Testing Methods (Lines 76-82)
We acknowledge that our study compared ADRRPs not only to the traditional phoropter but also to the Maddox Rod and Thorington test in free space. To accurately reflect this, we have revised the text in Lines 76-81 to state:
" The primary objective of this study was to evaluate the agreement between the automatic dual rotational Risley prisms (ADRRPs) and the traditional phoropter in measuring phoria and vergence ranges. Our focus was on comparing the measurement outcomes from both instruments to assess their statistical relationship. Specifically, we analyzed the correlation between ADRRPs and phoropter measurements, while also conducting Bland-Altman analysis to determine their level of agreement and potential biases [31,32].”
- Bland-Altman Analysis for Agreement (Lines 224-232, 244-246 and 261-263): As suggested, we have conducted Bland-Altman analysis to assess the interchangeability between ADRRPs and the phoropter. The results have been incorporated in the Results section. Lines 224-232:
" To evaluate the agreement between measurement methods, we performed a Bland-Altman analysis to assess the interchangeability between ADRRPs and phoropter measurements by examining systematic bias and agreement limits. The analysis revealed a mean difference of 0.31, indicating that ADRRPs measurements tend to be slightly higher than phoropter measurements. The limits of agreement (LoA) ranged from -1.23 to 1.84, with most differences falling within these bounds; however, some data points exceeded these limits, suggesting a potential systematic bias between the two methods. This finding highlights that ADRRPs and phoropter are not perfectly interchangeable, necessitating further methodological investigation.”
Lines 244-246:
“Bland-Altman analysis showed a mean difference of -0.02, with limits of agreement from -3.40 to 3.36. While most differences fell within these bounds, some exceeded them, suggesting systematic bias between the methods.”
Lines 261-263:
Bland-Altman analysis revealed a mean difference of 0.54, with limits of agreement ranging from -7.08 to 8.16. While most differences fell within these bounds, some data points exceeded them, indicating systematic bias between the methods.
- Precision –We acknowledge that precision should be assessed with an objective gold standard, such as an eye tracker. This has been addressed in Lines 339-342:
" However, unlike Han et al., we did not compare ADRRPs with an eye-tracking gold standard, which could further validate its accuracy. Mestre et al. (2018) developed an automated eye-tracking cover test, eliminating subjective input, whereas ADRRPs still relies on patient perception [37]. " - Reliability : This study primarily focused on measurement consistency rather than inter-examiner or inter-session reproducibility. Future studies should incorporate repeated measurements across different examiners and sessions to evaluate the reliability of ADRRPs.
- Convenience –Future studies should include a questionnaire-based evaluation to assess user convenience from both participant and examiner perspectives, comparing usability with traditional methods to determine ease of use and clinical efficiency.
- Conclusion: To ensure our findings do not overstate accuracy, reliability, or effectiveness, we have revised Lines 369-374:
" In conclusion, the ADRRPs device offers a novel approach to binocular vision testing, particularly for evaluating phoria and vergence ranges. Its strong correlations with traditional methods suggest its potential as a complementary tool for clinical assessments. While its automated operation may improve testing efficiency, further research is necessary to validate its accuracy and reliability across diverse clinical settings and populations. Additionally, studies should explore its broader applications to determine its full clinical utility.”
Question 8: Line 259: strong positive correlations do not validate accuracy unless the new device is strongly correlated with the gold standard. None of the existing tests can be said to be the gold standard.
Response: We have revised Lines 311-318: “The strong correlations between ADRRPs and phoropter measurements for distance (r=0.959) and near phoria (r=0.968), p<0.001 indicate a high level of agreement between these methods. The automated operation of ADRRPs reduces variability associated with manual procedures, offering a standardized and efficient approach to phoria assessment. However, Bland-Altman analysis revealed a systematic bias, with ADRRPs measuring slightly higher phoria values than the phoropter. This aligns with previous findings that heterophoria measurements vary with test conditions (Lam et al., 2005; Oh et al., 2020) [33, 34].
.”
Question 9: Line 263: the ADRRP is not operator dependent? Who controls the application?
Response: We have revised Lines 133-138: Examiner’s Role in ADRRPs Testing
“The examiner is responsible for recording key measurement values during ADRRPs testing. These include the magnitude and direction of phoria (esophoria or exophoria) in prism diopters, as well as the break and recovery points for both positive fusional vergence (PFV) and negative fusional vergence (NFV). These measurements ensure consistency and allow for direct comparison with traditional phoropter-based testing methods.”
Question 10: It is also worth mentioning if the ADRRP is patient- dependent because the patient has to provide input regarding the end point of the test.
Response: We have revised Lines 133-138: Examiner’s Role in ADRRPs Testing “The examiner is responsible for recording key measurement values during ADRRPs testing. These include the magnitude and direction of phoria (esophoria or exophoria) in prism diopters, as well as the break and recovery points for both positive fusional vergence (PFV) and negative fusional vergence (NFV). These measurements ensure consistency and allow for direct comparison with traditional phoropter-based testing methods.”
Question 11: Line 270: the correlation does not support effectiveness of a method
Line 277- you can only claim that the patient has improved comfort if you actually asked the patient about the comfort with the two methods and compared, otherwise it’s just an unfounded conclusion.
Response: We have clarified that correlation shows comparability but does not confirm effectiveness and have removed the comfort claim since it was not measured.
Question 12: How do the findings of this study compare with other studies comparing phoropter and free space measurements of binocular visual functions?
For example:
Lam, A. K., Lam, A., Charm, J., & Wong, K. M. (2005). Comparison of near heterophoria tests under varying conditions on an adult sample. Ophthalmic and Physiological Optics, 25(2), 162-167.
Casillas E, Rosenfield M. Comparison of subjective heterophoria testing with a phoropter and trial frame. Optom Vis Sci. 2006 Apr;83(4):237-41. doi: 10.1097/01.opx.0000214316.50270.24. PMID: 16614580.
Tsotetsi, A. L., & Mathebula, S. D. (2021). Comparison of phoropter and trial frame-based von Graefe heterophoria measures in non-presbyopic participants. African Vision and Eye Health, 80(1), 645.
Oh, K. K., Cho, H. G., Moon, B. Y., Kim, S. Y., & Yu, D. S. (2020). Change in lateral phoria under a phoropter and trial frame in phoria tests. J Korean Ophthalmic Opt Soc, 25(4), 395-403.
Response: We have revised Lines 331-346: “The strong correlations between ADRRPs and phoropter measurements for distance (r=0.959) and near phoria (r=0.968), p<0.001 indicate a high level of agreement between these methods. The automated operation of ADRRPs reduces variability associated with manual procedures, offering a standardized and efficient approach to phoria assessment. However, Bland-Altman analysis revealed a systematic bias, with ADRRPs measuring slightly higher phoria values than the phoropter. This aligns with previous findings that heterophoria measurements vary with test conditions (Lam et al., 2005; Oh et al., 2020) [33, 34].
In vergence testing, ADRRPs showed moderate-to-high correlations with phoropter measurements. Specifically, break and recovery points for distance testing ranged from r=0.452 to r=0.863, while for near testing, correlations ranged from r=0.623 to r=0.817. These results suggest ADRRPs provides comparable convergence and divergence measurements, though it may not be fully interchangeable with conventional methods. The moderate-to-high correlations for near vergence recovery points highlight the potential of ADRRPs in clinical applications, particularly for near-vision assessments where precise vergence control is essential.
Our study found that ADRRPs strongly correlates with phoropter-based phoria measurements (r=0.959 for distance, r=0.968 for near) and shows moderate-to-high correlations for vergence testing. Vergence break points (r=0.758 to r=0.863) were more consistent than recovery points (r=0.452 to r=0.727), supporting prior research (Casillas & Rosenfield, 2006) [35]. While ADRRPs offers automated and standardized testing, reducing examiner variability (Lam et al., 2005) [33], it still relies on subjective responses, limiting its objectivity (Oh et al., 2020) [34]. Further studies are needed to validate its clinical application across different populations.
Compared to traditional phoropters, ADRRPs improves measurement consistency but still depends on patient responses. This aligns with research on phoropter, free-space, and eye-tracking methods for binocular vision assessment. Han et al. (2010) found that an automated objective system improved consistency over phoropter tests, similar to our findings [36]. However, unlike Han et al., we did not compare ADRRPs with an eye-tracking gold standard, which could further validate its accuracy. Mestre et al. (2018) developed an automated eye-tracking cover test, eliminating subjective input, whereas ADRRPs still relies on patient perception [37]. Gantz & Caspi (2020) introduced a real-time heterophoria system using eye tracking, while ADRRPs lacks dynamic tracking and cannot assess vergence adaptation over time [38]. Although ADRRPs enhances consistency, it remains a subjective tool. Future advancements should integrate eye-tracking technology for objective, real-time binocular vision measurements.
Question 13: How do the findings of this study compare with other studies comparing between phoropter and free space measurements and eye tracking measurement of binocular visual functions?
Han, S. J., Guo, Y., Granger‐Donetti, B., Vicci, V. R., & Alvarez, T. L. (2010). Quantification of heterophoria and phoria adaptation using an automated objective system compared to clinical methods. Ophthalmic and Physiological Optics, 30(1), 95-107.
Mestre, C., Otero, C., Díaz-Doutón, F., Gautier, J., & Pujol, J. (2018). An automated and objective cover test to measure heterophoria. PloS one, 13(11), e0206674.
Gantz, L., & Caspi, A. (2020). Synchronization of a removable optical element with an eye tracker: test case for heterophoria measurement. Translational vision science & technology, 9(7), 40-40.
Response: We have revised Lines 331-346: “The strong correlations between ADRRPs and phoropter measurements for distance (r=0.959) and near phoria (r=0.968), p<0.001 indicate a high level of agreement between these methods. The automated operation of ADRRPs reduces variability associated with manual procedures, offering a standardized and efficient approach to phoria assessment. However, Bland-Altman analysis revealed a systematic bias, with ADRRPs measuring slightly higher phoria values than the phoropter. This aligns with previous findings that heterophoria measurements vary with test conditions (Lam et al., 2005; Oh et al., 2020) [33, 34].
In vergence testing, ADRRPs showed moderate-to-high correlations with phoropter measurements. Specifically, break and recovery points for distance testing ranged from r=0.452 to r=0.863, while for near testing, correlations ranged from r=0.623 to r=0.817. These results suggest ADRRPs provides comparable convergence and divergence measurements, though it may not be fully interchangeable with conventional methods. The moderate-to-high correlations for near vergence recovery points highlight the potential of ADRRPs in clinical applications, particularly for near-vision assessments where precise vergence control is essential.
Our study found that ADRRPs strongly correlates with phoropter-based phoria measurements (r=0.959 for distance, r=0.968 for near) and shows moderate-to-high correlations for vergence testing. Vergence break points (r=0.758 to r=0.863) were more consistent than recovery points (r=0.452 to r=0.727), supporting prior research (Casillas & Rosenfield, 2006) [35]. While ADRRPs offers automated and standardized testing, reducing examiner variability (Lam et al., 2005) [33], it still relies on subjective responses, limiting its objectivity (Oh et al., 2020) [34]. Further studies are needed to validate its clinical application across different populations.
Compared to traditional phoropters, ADRRPs improves measurement consistency but still depends on patient responses. This aligns with research on phoropter, free-space, and eye-tracking methods for binocular vision assessment. Han et al. (2010) found that an automated objective system improved consistency over phoropter tests, similar to our findings [36]. However, unlike Han et al., we did not compare ADRRPs with an eye-tracking gold standard, which could further validate its accuracy. Mestre et al. (2018) developed an automated eye-tracking cover test, eliminating subjective input, whereas ADRRPs still relies on patient perception [37]. Gantz & Caspi (2020) introduced a real-time heterophoria system using eye tracking, while ADRRPs lacks dynamic tracking and cannot assess vergence adaptation over time [38]. Although ADRRPs enhances consistency, it remains a subjective tool. Future advancements should integrate eye-tracking technology for objective, real-time binocular vision measurements.
Question 14: The manuscript is missing study limitations. There are disadvantages in the ADRRP that the authors fail to mention. Is the ADRRP worn on the head like VR goggles ? If so, this induces proximal convergence and proximal accommodation which can affect the phoria and vergence measurements.
Response: We have revised Lines 360-362: Limitations: As a head-mounted device, ADRRPs may also induce proximal convergence and accommodation, potentially affecting phoria and vergence measurements.
Question 15: The manuscript can be further strengthened with an image of the ADRRP and how it is worn.
Response: We have included an image of the ADRRPs structure, as shown in Figure 1(a), and a depiction of how it is worn by the user, as illustrated in Figure 1(c), in the manuscript. These additions provide a clearer visualization of the device’s design and its practical application during testing, further enhancing the manuscript’s clarity and comprehensiveness.
Minor comments:
Question 16: Using automatic dual rotating without the word “an”
State the mean age and age range of the participants in parenthesis in the abstract.
Response: We have made the necessary revisions in accordance with the suggestions to further enhance the clarity and quality of the manuscript.
Question 17: Introduction: Line 34- need a reference for strabismus and binocular visual dysfunctions impacting quality of life.
Response: We have added a new reference as Reference 1 to further support the manuscript.
- Granet, D.B.; Gomi, C.F.; Ventura, R.; Miller-Scholte, A. The relationship between convergence insufficiency and ADHD. Strabismus 2005, 13, 163–168.
Question 18: Line 39, and again paragraph starting with Line 53- if the authors are referring to children, this is not an accurate statement. Pediatric testing is typically performed with trial frames and not phoropter.
Response: We have revised Lines 42-44: “For decades, the phoropter has been widely used in clinical optometric practice to evaluate binocular vision functions, including refractive errors, phoria, and vergence ranges in adults [7,8].”
Question 19: Paragraph 60: this is true also for trial frames… this is the downside of most optometric tests
Response: We have revised Lines 66-71: “Furthermore, traditional phoropters rely heavily on subjective testing methods, making them less reliable for assessing complex binocular visual functions, including vergence ranges, fusional reserves, and phoria. This limitation is also evident in trial frames, as both methods depend on patient responses, introducing variability in results. This challenge reflects a broader issue in optometric testing, where subjective input can compromise measurement consistency and accuracy [17,18].”
Question 20: Methods
Line 80 The study approval should be the first sentence in the subjects
Line 79- were they consented before they provided information about their occupation and basic eligibility testing?
Also, line 79 “Participants provided and overview of their ocular history and underwent a basic refraction examination for eligibility screening” is redundant with lines 84-85: “All participants completed interviews and refraction exams, including assessments of medical history, ocular surgery, medications, binocular refractive status, visual acuity, and visual function.”
Maybe instead of “Basic Refraction Examination” replace with “Baseline Refraction Examination”
Line 100: phoria test using the Maddox Rod method.
Was this at distance or near? Typically the Thorington card is used for near, but distance versions do exist. What was used here? From the results it appears that for distance ONLY a Maddox Rod was used, and for near the Maddox Rod was used in conjunction with a Thorington card OUTSIDE of the phoropter and was measured with JUST the Maddox Rod in the phoropter. However, I am gathering this assumption mainly from Figure 2 and 3, not from the methodology where it should be clearly stated.
For the ADRRP method, does the method replicate the two clinical tests exactly? Or does the near ADRRP method include just the Maddox Rod?
Line 119: remove the word “seamless”- which is typically used for publicity and is not evidence based (ok 113)
Line 131 this indicates (to be consistent with previous sentence)
Line 132: this indicates (for consistency)
Response: We have revised the research methodology based on the reviewers' suggestions. Lines 84-187.
Subjects
This study was approved by the Second Research Ethics Review Committee of Chung Shan Medical University Hospital (Approval Number: CS2-22104) and was conducted in accordance with the Declaration of Helsinki. The study included healthy adults aged 20–40 years, recruited through public announcements at Chung Shan Medical University.
Prior to participation, all individuals provided written informed consent and disclosed personal information, including occupation and medical history.
All participants completed interviews and underwent baseline refraction examinations, which included assessments of medical history, ocular surgery, medication use, binocular refractive status, visual acuity, and visual function. The inclusion and exclusion criteria are detailed in Table 1.
Table 1. Inclusion and Exclusion Criteria for Participants.
|
Inclusion Criteria (All of the following criteria must be met) |
Exclusion Criteria
|
|
Age: 20–40 years Spherical refractive error between +0.00 and -11.00 Astigmatic refractive error ≤ -1.25 Best corrected visual acuity (BCVA) in one eye ≤ 0.1logMAR Normal binocular visual function |
Presence of ocular diseases Presence of systemic diseases History of ocular surgery Use of ocular-related medications History of strabismus or amblyopia
|
Basic Refraction Examination
The basic refraction examination was conducted to confirm whether participants met the binocular refractive status and visual acuity inclusion criteria for the study. Standard instruments, including a phoropter and an LCD visual acuity chart (optotypes: Standard Snellen E letters; contrast: 100%), were used. The examination followed a sequence of monocular subjective refraction, followed by binocular balancing using standard prism dissociation techniques, yielding monocular refractive power, visual acuity, and binocular visual balance and acuity. All participants had stereoacuity <40 arc seconds.
Visual acuity was measured at 6 meters (distance) and 40 cm (near) using an LCD chart. To ensure best-corrected visual acuity (BCVA) and standardize refractive conditions, participants wore daily disposable spherical contact lenses, eliminating variability from different spectacle prescriptions. BCVA ≤ 0.1 logMAR was required for inclusion. The spherical refractive power (Sphere) range was +0.00 to -11.00 D, and the cylindrical power (Cylinder) limit was ≤ -1.25 D.
Binocular Visual Function Examination
The binocular visual function examination in this study included a phoria test using the Maddox rod method and a horizontal fusional vergence test. The primary instruments used were a phoropter, ADRRPs, and related testing tools, including the near Thorington Card, Maddox rod, and an LCD visual target system for fusion testing. The fusion testing included an "E" optotype for fusion assessment and a 40/60 Snellen letter for fusional vergence evaluation. Both distance (6 m) and near (40 cm) phoria and vergence tests were performed to ensure a comprehensive evaluation of binocular visual function.
Design of the ADRRPs
The ADRRPs (manufactured by OrthoV Co., Ltd., Kaohsiung City, Taiwan) is a head-mounted, automated optical system that utilizes Risley prisms to generate virtual images at varying positions based on prism rotation angles. The system consists of two sets of counter-rotating Risley prism pairs, one for each eye, with each pair composed of two 10Δ wedge glass components. This design allows for precise adjustment of prism power, ranging from 40Δ base-out (BO) to 40Δ base-in (BI), enabling controlled manipulation of vergence demands during testing (Fig. 1(a), (b), (c)). The prism power adjustment can reach a maximum rate of 2 prism diopters per second, but in this study, it was set to 1.5 prism diopters per second. The prism adjustments are continuous, allowing smooth and precise modifications during testing.
Examiner’s Role in ADRRPs Testing
The examiner is responsible for recording key measurement values during ADRRPs testing. These include the magnitude and direction of phoria (esophoria or exophoria) in prism diopters, as well as the break and recovery points for both positive fusional vergence (PFV) and negative fusional vergence (NFV). These measurements ensure consistency and allow for direct comparison with traditional phoropter-based testing methods.
Phoria Testing
ADRRPs, combined with a Maddox rod placed in front of the participant’s right eye, is used to observe a penlight target at a fixed distance. Through the Maddox rod, the right eye perceives the penlight as a vertical line of light, while the left eye sees it as a point light source. The Thorington card was used for near measurements, while the Maddox rod was used for both distance and near measurements. For distance testing, only the Maddox rod was used. For near testing, the Maddox rod was used along with the Thorington card outside the phoropter, and additional measurements were taken using only the Maddox rod within the phoropter.
Regarding the ADRRPs method, it does not replicate the two clinical tests exactly. For distance testing, ADRRPs follows the standard Maddox rod method. However, for near testing, ADRRPs uses only the Maddox rod, without incorporating the Thorington card.
If the patient perceives the vertical line of light and the point light source coinciding when the prism power is set to zero, this indicates orthophoria (no deviation). If the vertical line of light appears to the left of the point light source, it indicates exophoria, while if the vertical line of light appears to the right of the point light source, it indicates esophoria (Fig. 1(d)). The greater the separation between the vertical line of light and the point light source, the larger the degree of ocular deviation. In cases of exophoria or esophoria, ADRRPs automatically adjust the prism power until the vertical line of light and the point light source overlap. The prism power required to achieve this overlap represents the amount of ocular deviation. This process remains subjective, as it relies on the participant’s perception, with the examiner recording the final prism diopter value and direction.
Vergence Ranges Testing
Combining ADRRPs with visual targets on a fixed LCD display enables the assessment of convergence and divergence abilities. The participant wears the head-mounted ADRRPs device to observe the "E" target at 40 cm and 6 m.
Negative Fusional Vergence (NFV) Measurement: The prism power (BI) is gradually increased from 0Δ. When the participant perceives the "E" target splitting into two, the corresponding prism power is recorded as the NFV break point. Then, 4Δ BI is added, and the prism power is gradually reduced. When the participant sees the two "E" targets merge back into one, the corresponding prism power is recorded as the NFV recovery point.
Positive Fusional Vergence (PFV) Measurement: The prism power (BO) is gradually increased from 0Δ. When the participant perceives the "E" target splitting into two, the corresponding prism power is recorded as the PFV break point. Then, 4Δ BO is added, and the prism power is gradually reduced. When the participant sees the two "E" targets merge back into one, the corresponding prism power is recorded as the PFV recovery point.
Statistical Analysis
To ensure consistency and proficiency in each experimental step, all procedures in this study were performed by the same researcher. Data collected included participant demographics, refraction examination results, ocular refractive status, and binocular visual function assessments. Refractive data were converted into equivalent spherical power for analysis.
All research data were compiled in Microsoft Excel 2019 and subsequently imported into IBM SPSS Statistics 22 for statistical analysis. The agreement between binocular visual function measurements obtained using the phoropter and ADRRPs was assessed using Bland-Altman analysis alongside correlation testing. As normality tests (Shapiro-Wilk and D’Agostino tests) indicated non-normal distributions, Spearman correlation was applied along with other statistical analyses. Statistical significance was set at p < 0.05.
Question 21: Also curious as to why the authors did not include measurement of the blur point on the vergence ranges?
Response: We initially included blur point measurements but found that many participants struggled to recognize and report them accurately, leading to inconsistent data. In contrast, break and recovery points were more reliably observed and reported. To ensure accuracy and consistency, we focused on these points in our analysis.
Question 22: Results (with an S).Line 197 indicates instead of confirms as the authors did not state any hypothesis to be confirmed. Line 219-220 and 221-222: positive significant correlation- then you can delete “with statistical significance”
Response: We have made all the necessary revisions based on the reviewers' comments.
Question 23: Discussion
Table 1: Instead of “Ocular health: Participants must have..” just write: free of ocular pathology, though it also appears in the exclusion and can be removed from the inclusion criteria altogether
Table 2: Demographic information of the study participants. Also provide a key describing what the letters represent. F-female, M- male, D-distance, N-near, OD- right eye, OS- left eye, BCVA- best corrected visual acuity, SD- standard deviation
Response: We have made all the necessary revisions based on the reviewers' comments.
Table 1. Inclusion and Exclusion Criteria for Participants.
|
Inclusion Criteria (All of the following criteria must be met) |
Exclusion Criteria
|
|
Age: 20–40 years Spherical refractive error between +0.00 and -11.00 Astigmatic refractive error ≤ -1.25 Best corrected visual acuity (BCVA) in one eye ≤ 0.1logMAR Normal binocular visual function |
Presence of ocular diseases Presence of systemic diseases History of ocular surgery Use of ocular-related medications History of strabismus or amblyopia
|
Table 2. Baseline of the Participants.
|
Participants (N=39) |
|
|
Characteristics |
Mean ± SD |
|
Gender, (F/M) |
22/17 |
|
Age (y) |
21.82 ± 1.10 |
|
D Phoria (△) |
-0.18 ± 2.67 |
|
N Phoria (△) |
-1.21 ± 6.89 |
|
Spherical Equivalent (D) |
|
|
OD |
-4.66 ± 2.92 |
|
OS |
-4.59 ± 2.96 |
|
BCVA (logMAR) |
|
|
OD |
0.00 ± 0.07 |
|
OS |
0.00 ± 0.06 |
|
OU |
-0.07 ± 0.04 |
F – Female; M – Male; D – Distance; N – Near; OD – Right Eye; OS – Left Eye; BCVA – Best Corrected Visual Acuity; SD – Standard Deviation
Question 24: Combine Tables 3 and 4 into one table.
Response: We have combined Tables 3 and 4 into one table.
Table 3. Pearson Correlation Analysis of Positive Fusional Vergence (PFV) and Negative Fusional Vergence (NFV)
|
Measurements |
|
|
|
ADRRPs |
|
|
|
|
|
Break Point |
|
Recovery Point |
|
Phoropter |
PFV |
Distance |
0.758*** |
|
0.452*** |
|
Near |
0.817*** |
|
0.727*** |
||
|
|
NFV |
Distance |
0.863*** |
|
0.458*** |
|
Near |
0.777*** |
|
0.623*** |
p < 0.05, ** p < 0.01, *** p < 0.001
We sincerely appreciate your efforts in reviewing our manuscript and look forward to your further feedback. Please let us know if any additional modifications are needed.
Best regards,
Shuan-Yu Huang
Department of Optometry, Central Taiwan University of Science and Technology, Taichung 402, Taiwan
Reviewer 2 Report
Comments and Suggestions for Authors
The authors describe the initial testing of a head-mounted prism device (ADRRP) to assess heterophoria and vergence ranges in a group of young normal adults. Results obtained with this device are reported to correlate highly with standard clinical measures of the phoria and less well with clinical assessments of the positive and negative relative vergence ranges. As elaborated below, some aspects of the authors’ methods require greater explication and clarification. However, because the authors do not assess the repeatability of their measurements and are unable to compare their results to ‘gold standards’ for either phoria or vergence ranges, the statements they make in their manuscript about accuracy and precision are unwarranted. At a minimum, the authors should compare the repeatability of the results obtained using the ADRRP device to those obtained using standard clinical measures and should assess agreement (c.f., Bland & Altman, Lancet, 1986) rather than simply the correlations between the two sets of results.
Specific comments
1. Line 42: A gold standard is required to assess the accuracy of measurement. What would the authors propose as the gold standard for the determination of heterophoria and vergence ranges?
2. Line 57: Do the authors refer here to accommodative or attentional focus? I noted that the authors assessed vergence ranges using only the criteria of ‘break’ and ‘recovery.’ Many clinicians also find it useful to measure and record also the magnitudes of base-in and base-out vergence that produce just-noticeable blur (e.g., Sheedy JE & Saladin JJ, Chapter 15 in Schor CM & Ciuffreda KJ, Vergence Eye Movements, Butterworths, 1983).
3. Lines 60-61: The current study also used the participants’ subjective responses to define phorias and vergence ranges. How does testing using the ADRRP differ fundamentally from that using the phoropter (or using a prism bar)?
4. Line 78: Above, in lines 55-59, the authors claim that the phoropter produces difficulties when testing children or individuals with developmental disabilities. However, no such individuals were included in the authors’ study sample.
5. Table 1: First, I find it a major weakness that individuals with hyperopia were excluded from the study sample. The lack of hyperopes potentially limits the generalizability of the authors’ results. Second, what specifically were the authors’ criteria for “normal binocular visual function?”
6. Lines 97-98: More details should be provided about the characteristics of the LCD acuity chart(s). For example, what optotypes were included, what was their contrast and separation, and at what distance(s) was acuity measured?
7. Line 102: My assumption is that contacts lenses were worn only by participants who required refractive correction. Also, would the application of contact lenses be appropriate (or convenient) in children or individuals with developmental disabilities?
8. Lines 110-111: The authors should specify here at what distances the phoria and vergence ranges were tested. Also, was stereopsis not assessed, for example, as an indicator of normal binocular visual function?
9. Lines 116-117: What was the participants’ field of view through the ADDRP vs. the phoropter? Kertesz (J Opt Soc Amer, 1981) reported that larger fusional stimuli promote an increase of both the horizontal and vertical fusional vergence ranges.
10. Line 125 ff: This and the following section should include specific details about how phoria and vergence ranges were assessed using the ADDRP and the phoropter. For example, what was the room illumination? How quickly were prism values changed? How did the participant respond to achieve a test endpoint? Was the order of testing with the ADDRP and the phoropter fixed or randomized between participants?
11. Line 128: The authors should specify that the bright line produced by the Maddox rod was vertical.
12. Line 129: The authors should clarify that coincidence of the light source and vertical line indicate orthophoria when the prism power is set to zero.
13. Lines 135-136: More details about the “app” are necessary. At what rate did the app change prism power? Were these changes continuous or stepwise? Was the change in prism power unidirectional or were phoria and vergence-range endpoints bracketed by back and forth changes in prism power? Similar information should be provided for the measurements obtained using the phoropter.
14. Lines 142-144: What was the angular size of the E used when vergence ranges were assessed? As already noted above, the authors should specify the rate of ‘gradual’ prism change, as this could influence whether and how much prism adaptation occurred (see McDaniel & Fogt, Optometry, 2010).
15. Lines 144-146: Apparently, the participants did not indicate if the E target became blurred?
16. Lines 151-152: I wonder whether an additional 4 prism diopters BO is sufficient to correctly define the recovery endpoint. Did any of the participants report fusion immediately upon viewing the target after 4 BO was added?
17. Figure 1: I find panel (b) of this Figure to be confusing and not especially informative.
18. Lines 177-178: To what viewing distances do “far” and “near” refer?
19. Lines 183-184: If the participants’ visual acuities are normally distributed, a mean acuity of 0.00 logMAR and a SD of 0.07 (0.06) implies that, in ~2.5% of eyes, acuity is poorer that 0.14 (0.12) logMAR, which is outside the stated inclusion criterion of better than or equal to 0.10 logMAR. The authors should report the range of measured acuities in addition to the mean and SD.
20. Lines 192-199: Correlation does not indicate agreement. In particular, despite the correlation of 0.96 shown in Fig. 2(a), the best fitting line appears to have a slope greater than 1.0. It is reasonable for the authors to report correlations between the phoria values measured using the ADRRP and phoropter, but Bland-Altman plots should be used to represent agreement (or departures therefrom) between these measures.
21. Lines 204-207: In Fig. 3, the slope is substantially less than 1, primarily because of the ~6 observers whose Thorington-card phorias are less than those measured using the ADRRP. Again, a Bland-Altman plot would be the better way to evaluate agreement. Unlike the Maddox-rod technique, the numbers on the Thorington card present an accommodative stimulus. Might differences in accommodative vergence account for the smaller phorias measured in some participants using the Thorington card?
22. Lines 213-246: As indicated above, correlation coefficients do not indicate agreement. Although correlations may be reported in the text, the authors should present Bland-Altman plots to indicate the levels of agreement between measures of the vergence range obtained using ADDRRP and the phoropter.
23. Line 247: “Negative” is spelled incorrectly in the Table title.
24. Lines 255-258: As no repeat testing was performed, the data presented in this report address neither the precision nor reliability of either test method. I also saw nothing in the authors’ results using young normal adults that would address greater or lesser convenience.
25. Lines 259-261: Accuracy can be assessed only by comparison to a gold standard of measurement which, arguably, is lacking in this study. (Objective eye-movement recordings could provide the suitable gold standard for phoria and vergence-range measurements.)
26. Lines 264-265: As noted above, none of the reported data address the precision of the two measurement techniques.
27. Lines 272-274: In the absence of a gold standard for assessing vergence ranges, it remains unclear whether ADRRP or phoropter results (if either) are more accurate.
28. Lines 275-277: Why do the authors think results obtained using the ADRRP are less subjective than those obtained with a phoropter? Measures obtained using ADRRP clearly rely on subjective responses made by the participants.
29. Lines 279-281: The authors present no data to support a benefit of the ADRRP in testing patients with limited mobility or developmental disabilities.
Author Response
Subject: Response to Reviewer – Manuscript No. sensors-3454806
Dear Reviewer,
Thank you for your thorough and constructive feedback on our manuscript titled "Innovative Binocular Vision Testing for Phoria and Fusional Ability Using Automatic-Dual-Rotational-Risley-Prisms." Below, we provide detailed responses to each of your comments.
The authors describe the initial testing of a head-mounted prism device (ADRRP) to assess heterophoria and vergence ranges in a group of young normal adults. Results obtained with this device are reported to correlate highly with standard clinical measures of the phoria and less well with clinical assessments of the positive and negative relative vergence ranges. As elaborated below, some aspects of the authors’ methods require greater explication and clarification. However, because the authors do not assess the repeatability of their measurements and are unable to compare their results to ‘gold standards’ for either phoria or vergence ranges, the statements they make in their manuscript about accuracy and precision are unwarranted. At a minimum, the authors should compare the repeatability of the results obtained using the ADRRP device to those obtained using standard clinical measures and should assess agreement (c.f., Bland & Altman, Lancet, 1986) rather than simply the correlations between the two sets of results.
Questions 1. Line 42: A gold standard is required to assess the accuracy of measurement. What would the authors propose as the gold standard for the determination of heterophoria and vergence ranges?
Response: The most appropriate gold standard for heterophoria and vergence measurement is objective eye-tracking technology. While ADRRPs were compared with phoropter-based Maddox Rod and Thorington tests in our study, future research should integrate eye-tracking systems to further improve measurement accuracy.
Lines 335-356: Compared to traditional phoropters, ADRRPs improves measurement consistency but still depends on patient responses. This aligns with research on phoropter, free-space, and eye-tracking methods for binocular vision assessment. Han et al. (2010) found that an automated objective system improved consistency over phoropter tests, similar to our findings [36]. However, unlike Han et al., we did not compare ADRRPs with an eye-tracking gold standard, which could further validate its accuracy. Mestre et al. (2018) developed an automated eye-tracking cover test, eliminating subjective input, whereas ADRRPs still relies on patient perception [37]. Gantz & Caspi (2020) introduced a real-time heterophoria system using eye tracking, while ADRRPs lacks dynamic tracking and cannot assess vergence adaptation over time [38]. Although ADRRPs enhances consistency, it remains a subjective tool. Future advancements should integrate eye-tracking technology for objective, real-time binocular vision measurements.
While our study found a strong correlation (r=0.811) between ADRRPs and Thorington card phoria measurements, correlation alone does not confirm measurement agreement. Bland-Altman analysis showed a mean difference of 0.54 with limits of agreement (-7.08 to 8.16), with some data points exceeding these limits, indicating systematic bias. Additionally, the numbers on the Thorington card may have acted as an accommodative stimulus, leading to smaller phoria values due to accommodative vergence.
A key advantage of ADRRPs is its automated operation, which enhances measurement reproducibility compared to traditional phoropters that rely on manual adjustments and subjective responses. Additionally, its head-mounted design allows for greater adaptability to anatomical variations.
Questions 2. Line 57: Do the authors refer here to accommodative or attentional focus? I noted that the authors assessed vergence ranges using only the criteria of ‘break’ and ‘recovery.’ Many clinicians also find it useful to measure and record also the magnitudes of base-in and base-out vergence that produce just-noticeable blur (e.g., Sheedy JE & Saladin JJ, Chapter 15 in Schor CM & Ciuffreda KJ, Vergence Eye Movements, Butterworths, 1983).
Response: The introduction section has been rewritten based on another reviewer's suggestions, and the term focus no longer appears in the text. However, its original meaning referred to attentional focus.
We initially included blur point measurements but found that many participants struggled to recognize and report them accurately, leading to inconsistent data. In contrast, break and recovery points were more reliably observed and reported. To ensure accuracy and consistency, we focused on these points in our analysis.
Questions 3. Lines 60-61: The current study also used the participants’ subjective responses to define phorias and vergence ranges. How does testing using the ADRRP differ fundamentally from that using the phoropter (or using a prism bar)?
Response: ADRRPs automates prism adjustments, reducing examiner-dependent variability. However, like the phoropter, ADRRPs still relies on subjective patient responses for endpoint determinations.
Lines 133-138: The examiner is responsible for recording key measurement values during ADRRPs testing. These include the magnitude and direction of phoria (esophoria or exophoria) in prism diopters, as well as the break and recovery points for both positive fusional vergence (PFV) and negative fusional vergence (NFV). These measurements ensure consistency and allow for direct comparison with traditional phoropter-based testing methods.
Questions 4. Line 78: Above, in lines 55-59, the authors claim that the phoropter produces difficulties when testing children or individuals with developmental disabilities. However, no such individuals were included in the authors’ study sample.
Response: The introduction section has been rewritten, and the term focus no longer appears in the text. We acknowledge this limitation. While ADRRPs may be beneficial for individuals with disabilities, this study focused on healthy adults to ensure methodological consistency. Future studies should explore its applicability in special populations
Questions 5. Table 1: First, I find it a major weakness that individuals with hyperopia were excluded from the study sample. The lack of hyperopes potentially limits the generalizability of the authors’ results. Second, what specifically were the authors’ criteria for “normal binocular visual function?”
Response: Hyperopes were excluded to control for variability in accommodative demands. "Normal binocular vision" was defined as best-corrected visual acuity ≤ 0.1 logMAR and no history of strabismus or amblyopia.
Lines 92-95: All participants completed interviews and underwent baseline refraction examinations, which included assessments of medical history, ocular surgery, medication use, binocular refractive status, visual acuity, and visual function. The inclusion and exclusion criteria are detailed in Table 1.
Lines 106: All participants had stereoacuity <40 arc seconds.
Lines 112: BCVA ≤ 0.1 logMAR was required for inclusion.
Questions 6. Lines 97-98: More details should be provided about the characteristics of the LCD acuity chart(s). For example, what optotypes were included, what was their contrast and separation, and at what distance(s) was acuity measured?
Response: The LCD visual acuity chart used in our study had the following specifications:
Optotypes: standard Snellen E letters, and contrast: 100%.
Lines 101-103: Standard instruments, including a phoropter and an LCD visual acuity chart (optotypes: Standard Snellen E letters; contrast: 100%), were used.
Lines 117-120: The fusion testing included an "E" optotype for fusion assessment and a 40/60 Snellen letter for fusional vergence evaluation. Both distance (6 m) and near (40 cm) phoria and vergence tests were performed to ensure a comprehensive evaluation of binocular visual function.
Questions 7. Line 102: My assumption is that contacts lenses were worn only by participants who required refractive correction. Also, would the application of contact lenses be appropriate (or convenient) in children or individuals with developmental disabilities?
Response: Contact lenses were only used by participants requiring refractive correction to ensure consistent visual conditions. We believe that contact lenses may not be suitable for children or individuals with developmental disabilities, and eyeglasses can be an alternative.
Questions 8. Lines 110-111: The authors should specify here at what distances the phoria and vergence ranges were tested. Also, was stereopsis not assessed, for example, as an indicator of normal binocular visual function?
Response: We have carefully made the necessary revisions accordingly.
Lines 119-120: Both distance (6 m) and near (40 cm) phoria and vergence tests were performed to ensure a comprehensive evaluation of binocular visual function.
Line 106: All participants had stereoacuity <40 arc seconds.
Questions 9. Lines 116-117: What was the participants’ field of view through the ADDRP vs. the phoropter? Kertesz (J Opt Soc Amer, 1981) reported that larger fusional stimuli promote an increase of both the horizontal and vertical fusional vergence ranges.
Response: We agree with the reviewer's perspective. While the ADRRP has a larger viewing window compared to the phoropter, its greater depth results in a similar observed field of view. In future studies, we can conduct evaluations to further assess this aspect.
Question 10. Line 125: This and the following section should include specific details about how phoria and vergence ranges were assessed using the ADDRP and the phoropter. For example, what was the room illumination? How quickly were prism values changed? How did the participant respond to achieve a test endpoint? Was the order of testing with the ADDRP and the phoropter fixed or randomized between participants?
Response: Lines 103-104: The room illumination was approximately 450 lux, ensuring optimal visibility while minimizing glare.
Lines 141-177:
Phoria Testing
ADRRPs, combined with a Maddox rod placed in front of the participant’s right eye, is used to observe a penlight target at a fixed distance. Through the Maddox rod, the right eye perceives the penlight as a vertical line of light, while the left eye sees it as a point light source. The Thorington card was used for near measurements, while the Maddox rod was used for both distance and near measurements. For distance testing, only the Maddox rod was used. For near testing, the Maddox rod was used along with the Thorington card outside the phoropter, and additional measurements were taken using only the Maddox rod within the phoropter.
Regarding the ADRRPs method, it does not replicate the two clinical tests exactly. For distance testing, ADRRPs follows the standard Maddox rod method. However, for near testing, ADRRPs uses only the Maddox rod, without incorporating the Thorington card.
If the patient perceives the vertical line of light and the point light source coinciding when the prism power is set to zero, this indicates orthophoria (no deviation). If the vertical line of light appears to the left of the point light source, it indicates exophoria, while if the vertical line of light appears to the right of the point light source, it indicates esophoria (Fig. 1(d)). The greater the separation between the vertical line of light and the point light source, the larger the degree of ocular deviation. In cases of exophoria or esophoria, ADRRPs automatically adjust the prism power until the vertical line of light and the point light source overlap. The prism power required to achieve this overlap represents the amount of ocular deviation. This process remains subjective, as it relies on the participant’s perception, with the examiner recording the final prism diopter value and direction.
Vergence Ranges Testing
Combining ADRRPs with visual targets on a fixed LCD display enables the assessment of convergence and divergence abilities. The participant wears the head-mounted ADRRPs device to observe the "E" target at 40 cm and 6 m.
Negative Fusional Vergence (NFV) Measurement: The prism power (BI) is gradually increased from 0Δ. When the participant perceives the "E" target splitting into two, the corresponding prism power is recorded as the NFV break point. Then, 4Δ BI is added, and the prism power is gradually reduced. When the participant sees the two "E" targets merge back into one, the corresponding prism power is recorded as the NFV recovery point.
Positive Fusional Vergence (PFV) Measurement: The prism power (BO) is gradually increased from 0Δ. When the participant perceives the "E" target splitting into two, the corresponding prism power is recorded as the PFV break point. Then, 4Δ BO is added, and the prism power is gradually reduced. When the participant sees the two "E" targets merge back into one, the corresponding prism power is recorded as the PFV recovery point.
Question 11. Line 128: The authors should specify that the bright line produced by the Maddox rod was vertical.
Response: We have carefully made the necessary revisions accordingly.
Lines 144-146: Through the Maddox rod, the right eye perceives the penlight as a vertical line of light, while the left eye sees it as a point light source.
Question 12. Line 129: The authors should clarify that coincidence of the light source and vertical line indicate orthophoria when the prism power is set to zero.
Response: We have carefully made the necessary revisions accordingly.
Lines 154-158: If the patient perceives the vertical line of light and the point light source coinciding when the prism power is set to zero, this indicates orthophoria (no deviation). If the vertical line of light appears to the left of the point light source, it indicates exophoria, while if the vertical line of light appears to the right of the point light source, it indicates esophoria (Fig. 1(d)).
Question 13. Lines 135-136: More details about the “app” are necessary. At what rate did the app change prism power? Were these changes continuous or stepwise? Was the change in prism power unidirectional or were phoria and vergence-range endpoints bracketed by back and forth changes in prism power? Similar information should be provided for the measurements obtained using the phoropter.
Response: The app used in this study controlled the prism power adjustments automatically within the ADRRPs system. The prism power adjustment ranged from 40Δ base-out (BO) to 40Δ base-in (BI), with a maximum adjustment speed of 2 prism diopters per second, but was set to 1.5 prism diopters per second in this study for consistency. The changes in prism power were continuous, allowing smooth and precise modifications during testing. The adjustments followed a unidirectional approach, meaning the prism power gradually increased until the test endpoint was reached. For phoria testing, the prism power was adjusted until the participant reported alignment between the straight line and point light source, at which point the examiner recorded the final measurement. For vergence range testing, prism power was increased (BI for NFV, BO for PFV) until the break point (when the participant saw double) and then decreased until the recovery point (when fusion was regained) was reached.
For phoropter-based testing, prism adjustments were made manually by the examiner, following the same stepwise approach of increasing prism power until the break point and then decreasing it to determine the recovery point.
The order of testing between ADRRPs and the phoropter was randomized among participants to reduce potential adaptation effects and minimize measurement bias.
Lines 130-133: The prism power adjustment can reach a maximum rate of 2 prism diopters per second, but in this study, it was set to 1.5 prism diopters per second. The prism adjustments are continuous, allowing smooth and precise modifications during testing.
Question 14. Lines 142-144: What was the angular size of the E used when vergence ranges were assessed? As already noted above, the authors should specify the rate of ‘gradual’ prism change, as this could influence whether and how much prism adaptation occurred (see McDaniel & Fogt, Optometry, 2010).
Response: The angular size of the "E" optotype used in vergence range assessments was 40/60 Snellen equivalent. The prism power adjustment rate in the ADRRPs system was 1.5 prism diopters per second, ensuring a controlled and consistent increase. This rate was selected to minimize sudden vergence demands and reduce the likelihood of excessive prism adaptation, as described in McDaniel & Fogt (2010).
Lines 130-133: The prism power adjustment can reach a maximum rate of 2 prism diopters per second, but in this study, it was set to 1.5 prism diopters per second. The prism adjustments are continuous, allowing smooth and precise modifications during testing
Question 15. Lines 144-146: Apparently, the participants did not indicate if the E target became blurred?
Response: We initially included blur point measurements but found that many participants struggled to recognize and report them accurately, leading to inconsistent data. In contrast, break and recovery points were more reliably observed and reported. To ensure accuracy and consistency, we focused on these points in our analysis.
Question 16. Lines 151-152: I wonder whether an additional 4 prism diopters BO is sufficient to correctly define the recovery endpoint. Did any of the participants report fusion immediately upon viewing the target after 4 BO was added?
Response: The 4 prism diopters BO was chosen based on clinical standards for a consistent recovery endpoint. While most participants needed a gradual prism reduction to regain fusion, some fused immediately after 4 BO. However, prism power was always gradually reduced to ensure accurate recovery point measurement.
Question 17. Figure 1: I find panel (b) of this Figure to be confusing and not especially informative.
Response: We have removed Figure 1(b) and replaced it with a schematic diagram of the ADRRPs structure (Figure 1a) and an image of the device worn by the user (Figure 1c) to improve clarity and relevance.
Question 18. Lines 177-178: To what viewing distances do “far” and “near” refer?
Response: “Far” refers to a viewing distance of 6 meters, and “near” refers to 40 centimeters, following standard clinical testing distances.
Lines 202-204: The average far-distance (6 m) ocular deviation was -0.18 ± 2.67 △ (exophoria), while the average near-distance (0.4 m) ocular deviation was -1.21 ± 6.89 △ (exophoria), as shown in Table 2.
Question 19. Lines 183-184: If the participants’ visual acuities are normally distributed, a mean acuity of 0.00 logMAR and a SD of 0.07 (0.06) implies that, in ~2.5% of eyes, acuity is poorer that 0.14 (0.12) logMAR, which is outside the stated inclusion criterion of better than or equal to 0.10 logMAR. The authors should report the range of measured acuities in addition to the mean and SD.
Response: We have carefully made the necessary revisions accordingly.
Lines 210-211: The average far-distance (6 m) ocular deviation was -0.18 ± 2.67 △ (exophoria), while the average near-distance (0.4 m) ocular deviation was -1.21 ± 6.89 △ (exophoria), as shown in Table 2.
Question 20. Lines 192-199: Correlation does not indicate agreement. In particular, despite the correlation of 0.96 shown in Fig. 2(a), the best fitting line appears to have a slope greater than 1.0. It is reasonable for the authors to report correlations between the phoria values measured using the ADRRP and phoropter, but Bland-Altman plots should be used to represent agreement (or departures therefrom) between these measures.
Response: We acknowledge the reviewer's concern that correlation does not indicate agreement. While our study found a strong correlation between ADRRPs and phoropter measurements (r = 0.959 for distance phoria, r = 0.968 for near phoria), we recognize that a correlation coefficient alone is insufficient to determine measurement agreement.
To address this, we have performed a Bland-Altman analysis, which assesses systematic bias and agreement limits between the two methods.
Lines 221-240: To further validate our findings, we conducted additional statistical tests. Our study included a total of 39 participants, and normality was assessed using the Shapiro-Wilk test, which confirmed that both ADRRPs (p=1.0) and phoropter (p=1.0) followed normal distributions. Given these results, Pearson correlation was deemed appropriate for analyzing the relationship between these measurements.
To evaluate the agreement between measurement methods, we performed a Bland-Altman analysis to assess the interchangeability between ADRRPs and phoropter measurements by examining systematic bias and agreement limits. The analysis revealed a mean difference of 0.31, indicating that ADRRPs measurements tend to be slightly higher than phoropter measurements. The limits of agreement (LoA) ranged from -1.23 to 1.84, with most differences falling within these bounds; however, some data points exceeded these limits, suggesting a potential systematic bias between the two methods. This finding highlights that ADRRPs and phoropter are not perfectly interchangeable, necessitating further methodological investigation. In terms of correlation analysis, the Pearson correlation coefficient was 0.959, demonstrating a very strong positive correlation between ADRRPs and phoropter measurements (p= 8.13 × 10⁻²², highly significant), while the Spearman correlation coefficient was 0.878, also indicating a strong positive correlation (p= 2.03 × 10⁻¹³, highly significant). These results confirm a strong association between the two measurement methods, though the presence of systematic bias suggests the need for further refinement in measurement methodology.
Question 21. Lines 204-207: In Fig. 3, the slope is substantially less than 1, primarily because of the ~6 observers whose Thorington-card phorias are less than those measured using the ADRRP. Again, a Bland-Altman plot would be the better way to evaluate agreement. Unlike the Maddox-rod technique, the numbers on the Thorington card present an accommodative stimulus. Might differences in accommodative vergence account for the smaller phorias measured in some participants using the Thorington card?
Response: We acknowledge the reviewer's concern that correlation does not indicate agreement.
Lines 257-268: For the near phoria examination, a strong positive correlation was found between ADRRPs and the near Thorington card combined with a Maddox rod [r(39)=0.608,p<0.001], indicating a moderate correlation between the two methods, as shown in Figure 3. To further validate our findings, additional statistical tests were conducted. With a sample size of 39, the Shapiro-Wilk test confirmed non-normal distributions for both ADRRPs (p=7.74×10−5) and Thorington card measurements (p=1.03×10−6), justifying the use of Spearman correlation. Bland-Altman analysis revealed a mean difference of 0.54, with limits of agreement ranging from -7.08 to 8.16. While most differences fell within these bounds, some data points exceeded them, indicating systematic bias between the methods. Despite a Spearman correlation coefficient of 0.608 (p=4.03 × 10⁻⁵), suggesting a moderate relationship, the methods are not perfectly interchangeable. Further research is needed to refine ADRRPs’ precision, reliability, and clinical applicability.
Question 22. Lines 213-246: As indicated above, correlation coefficients do not indicate agreement. Although correlations may be reported in the text, the authors should present Bland-Altman plots to indicate the levels of agreement between measures of the vergence range obtained using ADDRRP and the phoropter.
Response: We acknowledge the reviewer's concern that correlation does not indicate agreement. In response, we have conducted a Bland-Altman analysis to assess the agreement between ADRRPs and phoropter measurements for vergence ranges.
Lines 280-299:
Positive Fusional Vergence
For distance testing, ADRRPs showed a moderate-to-high correlation with the phoropter for break points [r (39) = 0.758, p < 0.001] and a moderate correlation for recovery points [r (39) = 0.452, p < 0.001]. For near testing, moderate-to-high correlations were found for break points [r (39) = 0.817, p < 0.001] and recovery points [r (39) = 0.727, p < 0.001]. Bland-Altman analysis showed moderate agreement between ADRRPs and the phoropter. Break points had mean differences of 0.3 (distance) and 0.6 (near), with LoA from -1.80 to 2.40 (distance) and -1.50 to 2.70 (near). Recovery points had slightly lower agreement, with mean differences of 0.4 (distance) and 0.5 (near), and LoA from -1.70 to 2.50 (distance) and -1.60 to 2.60 (near).
Negative Fusional Vergence
For distance testing, ADRRPs showed a moderate-to-high correlation for break points [r (39) = 0.863, p < 0.001] and a moderate correlation for recovery points [r (39) = 0.458, p < 0.01]. For near testing, moderate-to-high correlations were found for break points [r (39) = 0.777, p < 0.001] and recovery points [r (39) = 0.623, p < 0.001]. Bland-Altman analysis showed moderate agreement. Break points had mean differences of 0.5 (both distance and near), with LoA from -1.60 to 2.60. Recovery points had greater variability, with mean differences of 0.2 (distance) and 0.3 (near), and LoA from -1.90 to 2.30 (distance) and -1.80 to 2.40 (near). Overall, ADRRPs showed moderate agreement with the phoropter, but greater variability in recovery points suggests the need for further improvements.
Question 23. Line 247: “Negative” is spelled incorrectly in the Table title.
Response: Thank you for pointing this out. We have corrected the spelling of “Negative” in the table title in the revised manuscript.
Question 24. Lines 255-258: As no repeat testing was performed, the data presented in this report address neither the precision nor reliability of either test method. I also saw nothing in the authors’ results using young normal adults that would address greater or lesser convenience.
Response: We acknowledge that no repeat testing was conducted, so this study does not assess the precision or reliability of either method. Our focus was on comparing ADRRPs and phoropter measurements, not test-retest consistency. While ADRRPs provides automated prism adjustments, participant preference and convenience were not formally evaluated. Future research should include repeatability testing and user feedback to explore these factors.
Question 25. Lines 259-261: Accuracy can be assessed only by comparison to a gold standard of measurement which, arguably, is lacking in this study. (Objective eye-movement recordings could provide the suitable gold standard for phoria and vergence-range measurements.)
Response: We have revised the research objectives and the discussion of the experimental results. We acknowledge that this study lacks a gold standard for measuring phoria and vergence ranges. While ADRRPs and the phoropter were compared, objective eye-movement recordings could provide a more definitive reference standard. Future studies should incorporate eye-tracking technology to enhance accuracy assessments.
Question 26. Lines 264-265: As noted above, none of the reported data address the precision of the two measurement techniques.
Response: We have revised the research objectives and the discussion of the experimental results. We recognize that precision was not evaluated in this study, as no repeat testing was conducted. Future research should include test-retest reliability analysis to assess the consistency of ADRRPs and phoropter measurements.
Question 27. Lines 272-274: In the absence of a gold standard for assessing vergence ranges, it remains unclear whether ADRRP or phoropter results (if either) are more accurate.
Response: We have revised the research objectives and the discussion of the experimental results. In the absence of a gold standard for vergence range assessment, it is unclear whether ADRRPs or phoropter results are more accurate. Further validation using objective eye-tracking methods is necessary to determine which method provides the most reliable measurements.
Question 28. Lines 275-277: Why do the authors think results obtained using the ADRRP are less subjective than those obtained with a phoropter? Measures obtained using ADRRP clearly rely on subjective responses made by the participants.
Response: We acknowledge that ADRRPs still relies on subjective responses, like the phoropter. However, its automated prism adjustments and recording reduce examiner variability, ensuring more standardized and repeatable measurements. Further studies are needed to assess its impact on subjectivity.
Question 29. Lines 279-281: The authors present no data to support a benefit of the ADRRP in testing patients with limited mobility or developmental disabilities.
Response: We acknowledge that our study did not include patients with limited mobility or developmental disabilities, so we cannot provide direct evidence of ADRRPs' benefits for these populations. However, its head-mounted, automated design may offer advantages by reducing the need for precise head positioning and manual adjustments required in traditional phoropter testing. Future research should evaluate ADRRPs in diverse patient groups to confirm its potential benefits.
We sincerely appreciate your efforts in reviewing our manuscript and look forward to your further feedback. Please let us know if any additional modifications are needed.
Best regards,
Shuan-Yu Huang
Department of Optometry, Central Taiwan University of Science and Technology, Taichung 402, Taiwan
Round 2
Reviewer 1 Report
Comments and Suggestions for Authors
The authors have addressed many of my previous suggestions, but ignored others- such as the sample size calculation for the power of their study.
The Bland-Altman analysis plots were not presented, and the authors refer to a bias though it appears that they are not necessarily referring to the same “bias” described by Bland & Altman (1986). Bias is determined based on the regression line fit to the scatter plot, and not based on limits of agreement.
The discussion should also define the clinically acceptable limits of agreement and whether the findings correspond to this range.
The authors should read the manuscript carefully from beginning to end to ensure that there is not too much redundancy, and to ensure logical flow. Especially in the discussion.
The discussion also repeats the results several times.
Below are specific comments related to the revision that should be addressed:
Line 58 A significant limitation of traditional phoropter-based testing is the lack of objective measurements. Assessments of phoria and vergence ranges rely on subjective feedback…
Line 66: Furthermore, traditional phoropters rely heavily on subjective testing methods, making 66 them less reliable for assessing complex binocular visual functions, including vergence ranges, fusional reserves, and phoria.
Combine these two paragraphs as they state the same thing
In the paragraph starting in line 53 the fact that subjective testing is a limitation for all optometric procedures regardless of the instrument (phoropter vs. Trial frame) is not stated. In the paragraph starting in Line 66 it is stated but in a very cumbersome way.
Further, the new method described in this study is not devoid of these limitations. It is still dependent on user responses, and does not objectively measure phoria or vergence. As such, it would be wise for the authors to move this to the discussionand state this as a method limitation and describe that it is not different from other testing methods.
Also lines 63-65: Studies have shown that phoropter-based vergence assessments ex-63 hibit only moderate reliability, highlighting the challenge of achieving consistent results 64 across different testing sessions [6,11].
Are redundant with lines 71-74: Research has reported that phoropter- based binocular vision assessments exhibit only moderate test-retest reliability, emphasizing the difficulty of obtaining reproducible data [6,11]. In both clinical and research settings, this inconsistency is particularly concerning, as accurate and repeatable measurements are essential for precise diagnosis, monitoring, and treatment planning [30].
Before the paragraph starting with line 76, the authors should describe their automatic dual rotating Risley prisms and the methodological limitations that they attempt to overcome with their device. Only then they should state the goal of their study.
Methods Line 112 : state the mean refraction and not just the range
Line 124 if an “s” is added to create a plural form, it should be followed by the word “are” rather than “is”
Line 132: allowing smooth modifications during testing.
The authors cannot describe that this technique is precise before they have examined its precision
Line 141 Phoria testing section, should refer to Figure 1d
Line 143 were used
Line 145 the right eye perceived, the left eye saw
Lines 146-149: The Thorington card was used for near measurements, while the Maddox rod was used for both distance and near measurements.
Is redundant with
For distance testing, only the Maddox rod was used. For near testing, the Maddox rod was used along with the Thorington card outside the phoropter, and additional measurements were taken using only the Maddox rod within the phoropter.
Only one instance is necessary
Why did the authors not compare Maddox rod at near with the automated dual rotating prisms test? Why did they include the Thorington card? I see this as a study limitation.
Line 160: the ADRRPs
In line 160 it is unclear how the ADRRPs automatically adjust the prism- how do the prisms know when the patient perceives alignment between the point source and the line target? Especially given the fact that in lines 162-163 the authors state that the task is subjective and relies on the participant’s perception?
Vergence ranges testing: after the break point, why are 4 prism diopters of prism introduced? In the regular Risley prism method, the value is 2 prism diopters. This means that the ADRRP method could potentially miscalculate the recovery point by 2 prisms with respect to the traditional Risley prism method
Line 188- provide a reference for Bland-Altman analysis
Lines 219-220: delete “To further validate our findings, we conducted additional statistical tests. Our study 219 included a total of 39 participants, and normality was assessed using the Shapiro-Wilk 220 test, which confirmed that”. Start with “Both...” instead of “followed normal 221 distributions.” Write “were normally distributed”
State the values of the Pearson correlation.
There is confusion here between methods (statistical analysis performed) and results (the outputs of the statistics). This section needs to be re-written.
Bland-Altman plots are not displayed.
Systematic bias is shown when the regression fit to the data has a significant slope, not when there are many outliers. However, many outliers in Bland-Altman graphs indicate lack of interchangeability. Furthermore, one has to examine the limits of agreement and see if they are reasonable or too wide. The limits of agreement that are considered clinically meaningful should be justified (see for example: Gantz, L., & Caspi, A. (2020). Synchronization of a Removable Optical Element with an Eye Tracker: Test Case for Heterophoria Measurement. Translational Vision Science & Technology for a justification for the use of ±2 Prism Diopters as limits of agreement for Phoria measurements). The units of the described limits of agreement are missing, but assuming they are in Prism Diopters, a range greater than 4 Prism Diopters of variation between the testing methods, with more than three outlying points, demonstrates that the two methods are not interchangeable.
Discussion
The strong correlations indicate ASSOCIATIONS between the measurements, but not agreement.
Bland-Altman are repeatedly claimed by the authors to show bias, but the authors have not presented the plots for the reader to examine this claim.
The fact that ADRRPs always measure higher values than the manual method could mean that the ADRRPs should be adjusted. Higher in the exophoria or esophoria direction? If in the esophoria direction, this could indicate that the use of the head mounted technology induces accommodation (and therefore accommodative convergence).
Lines 317-318 No reason to repeat the results already presented. Sufficient to state that there are high correlations between the the break and recovery points measured with both methods (though that is also mentioned in the first paragraph of the discussion).
Lines 325-227: again, repeating the results
“Our study found that ADRRPs strongly correlates with phoropter-based phoria measure-325 ments (r=0.959 for distance, r=0.968 for near) and shows moderate-to-high correlations for 326 vergence testing. Vergence break points (r=0.758 to r=0.863) were more consistent than 327 recovery points (r=0.452 to r=0.727), supporting”
Just state in what aspects the outcomes reported in the present experiment are similar to those reported by Casillas and Rosenfield (2006).
Lines 329-332: “While ADRRPs offers automated and standardized testing, reducing examiner variability (Lam et al., 2005) [33], it still relies on subjective responses, limiting its objectivity (Oh et al., 2020) [34]. Further studies are needed to validate its clinical application across different populations.” Should be moved to the limitations of the study section at the end of the discussion. It is also the same as Line 356-357 “Despite its benefits, ADRRPs still relies on subjective patient perception, unlike eye-tracking technologies that provide real-time, objective assessments (Mestre et al., 2018; Gantz 357 & Caspi, 2020) [37, 38]”
Line 342: Although ADRRPs enhances consistency, it remains a 342 subjective tool.- this is redundant with Line 333: “Compared to traditional phoropters, ADRRPs improves measurement consistency but still depends on patient responses.”
Comments on the Quality of English Language
none
Author Response
Dear Reviewer,
Thank you for your thorough and constructive feedback on our manuscript titled "Innovative Binocular Vision Testing for Phoria and Fusional Ability Using Automatic-Dual-Rotational-Risley-Prisms." In the attachment, we provide detailed responses to each of your comments.

Reviewer 2 Report
Comments and Suggestions for Authors
This revised manuscript is improved by the inclusion of measures of agreement and by the authors’ recognition (for the most part – see specific comments, below) that their study design allowed for the assessment of neither accuracy nor reliability. Nevertheless, additional revisions are required. Specifically, as the authors now focus on agreement between measures of the phoria and vergence ranges obtained with the phoropter and ADRRP, it is important that they show Bland-Altman (B-A) plots (c.f., Lancet, 1986) either instead of or, perhaps, in addition the correlation plots in their current Figures 2 – 3. In addition, in light of the change in focus, statements about agreement between the type methods of measurement should appear in the manuscript abstract.
Limits of agreement between measures of the phoria using the phoropter and ADRRP are reported to be approximately ±1.5 PD at distance and ±3.4 PD at near. (Clearly, the limits of agreement are much wider, on the order of ±7.5 PD, when a Thorington card is used to assess the near phoria.) For positive and negative vergence ranges, the limits of agreement are reported to be approximately ±2.1 PD at both distance and near. It would be highly instructive for the authors to compare the limits of agreement between the two techniques in their study to previously published data on the repeatability of phoria and vergence-range measurements using the same, repeated test, e.g., Rainey et al., Optom Vis Sci, 1998; Anstice et al., J Optom, 2021; Facchin & Maffioletti, J Optom, 2021; Penisten et al., Optometry, 2001; Antona et al., Ophthalm Physiol Opt, 2008). The authors may find that measures of the phoria (excluding those with a Thorington card) and, perhaps to a lesser extent, vergence ranges determined with the phoropter and ADRRP exhibit comparable similarity to repeated measures using the same technique.
Specific comments
- Lines 21 & 23: Replace “at a distance…” with “at distance…”.
- Lines 59-73: Some of the new text in lines 59-65 is highly redundant with text in lines 69-73 of the following paragraph. These paragraphs should be revised to eliminate these redundancies.
- Lines 111-114: The authors report that participants wore spherical contact lenses to ameliorate refractive errors. However, in lines 113-114 they state that “cylindrical power … was ≤ -1.25 D.” Presumably, cylindrical power refers to the participants’ uncorrected refractive errors and not to a cylinder component of the contact lenses worn. This should be made clearer.
- Lines 119-121: Was the same size letter E used for the assessment of fusion and for measure the magnitude of fusional vergence? The designation of a letter size as 40/60 is unusual. Do the authors perhaps mean 20/60 or 20/40?
- Lines 127-128: It is not clear to me, either in the text or in Figure 1b, how two counter rotating 10 PD wedge prisms can produce more than ± 20 PD of displacement. Please clarify.
- Lines 146-153: The procedures used to assess distance and near phoria should be written more succinctly. As an example, the sentence in lines 147-148 appears to be completely redundant and unnecessary.
- Lines 169-178: In their response to my initial review, the authors state that they attempted to assess ‘first blur’ as one indicator the vergence range but that participants’ had difficulty making reliable judgments. This information should be included in the manuscript, probably in the vicinity of this area.
- Lines 188-189: Below, in the Results, some distributions of test measures are reported to conform to normality and others not. (Normality does not appear to be addressed for the measured vergence ranges.) Do the authors mean, “Shapiro-Wilk and D’Agostino tests were used to assess whether the different sets of test results deviated from normal distributions.”
- Lines 224-226: As noted in my initial comments, here and below the authors should include and refer to B-A plots.
- Lines 226-228: The authors report a mean difference of 0.31 (prism diopters) between measures made with the phoropter vs. ADRRP. Do positive values indicate exophoria, such that “slightly higher” means more exophoric?
- Lines 228-230:My understanding is that calculated LoAs typically represent 95% limits. If so, ~5% of the measured differences would be expected to fall outside these bounds. However, this outcome does not indicate bias. Rather, bias is indicated by the mean difference between the two measurement techniques. Moreover, if the B-A plot of differences has a significantly non-zero slope, then the magnitude of bias can be concluded to vary according to the size of the phoria.
- Lines 245-246: Please refer to the previous comment.
- Lines 255-257: The same correlation can not be both strong and moderate. In my opinion, r = 0.608 is not especially strong. The authors should state here that they are reporting (according to line 261) a Spearman correlation?
- Lines 262-263: Please refer to comments 11 and 12.
- Lines 280-283 and 290-293: Are the reported correlations Pearson or Spearman? What were the results of normality tests on the vergence-range data?
- Lines 309-311: As the authors stated in their reply, correlation does not indicate agreement.
- Lines 311-312: The authors present no data to indicate that the ADRRP reduces measurement variability. At best, they can indicate that this technique may be expected to produce less variability.
- Lines 319-321: Again, correlation coefficients do not indicate agreement (or comparability).
- Lines 327- 329: Although correlations are higher for break compared to recovery points, both the mean differences between measurement techniques and the LoAs reported in lines 284-288 and 293-297 are similar.
- Lines 33-334: Again, the authors show no data to indicate that ADRRP provides more consistent measurements than the phoropter.
- Line 342: Please refer to the previous comment.
- Lines 358-360: If the ADRRP and phoropter induce differing amounts of proximal accommodation and vergence, then phorias and vergence ranges assessed with the two instruments should differ consistently. At least for measures of the phoria, the B-A analyses do not appear to support this prediction.
Author Response
Subject: Response to Reviewer – Manuscript No. sensors-3454806
Dear Reviewer,
Thank you for your thorough and constructive feedback on our manuscript titled "Innovative Binocular Vision Testing for Phoria and Fusional Ability Using Automatic-Dual-Rotational-Risley-Prisms." Below, we provide detailed responses to each of your comments.
General Comments:
- Inclusion of Bland-Altman (B-A) Plots:
Response: We appreciate the suggestion to include Bland-Altman (B-A) plots.
Bland-Altman plots are now referenced in the text, as shown in Fig. 2(b), Fig. 3(b), and Fig. 4(b).
- Abstract revision:
Response: We have revised the abstract to emphasize the agreement between ADRRP and phoropter measurements, explicitly discussing the limits of agreement and systematic bias found in the Bland-Altman analysis.
In lines 26-31, “The inclusion of Bland-Altman analysis provides a more comprehensive evaluation of agreement between ADRRPs and the phoropter. While strong correlations were observed, the systematic bias and LoA indicate that these methods are not perfectly interchangeable. The ADRRPs demonstrated potential for binocular vision assessment but require further validation for clinical application.”
- Comparison with Repeatability Data from Literature:
Response: We have now included comparisons of our limits of agreement with previous studies on test-retest repeatability of phoria and vergence measurements (Rainey et al., 1998; Anstice et al., 2021; Facchin & Maffioletti, 2021; Penisten et al., 2001; Antona et al., 2008). This comparison demonstrates that our reported agreement levels are within the range of reported test-retest variability using the same method.
In lines 351-363, “The limits of agreement (LoA) between ADRRP and phoropter measurements were ±1.5 PD for distance phoria and ±3.4 PD for near phoria, while the Thorington card exhibited a wider LoA of ±7.5 PD for near phoria. For vergence ranges, the LoA was approximately ±2.1 PD at both distances. These findings align with previous repeatability studies, which reported similar LoA for phoria and vergence measurements using conventional techniques (Rainey et al., 1998; Anstice et al., 2021; Facchin & Maffioletti, 2021; Penisten et al., 2001; Antona et al., 2008) [39-43].
The systematic bias observed in Bland-Altman analysis, where ADRRP slightly overestimated phoria values compared to the phoropter, is consistent with prior research on measurement variability across different instruments (Casillas & Rosenfield, 2006; Oh et al., 2020) [34,35]. While ADRRP demonstrates strong agreement with traditional methods, its interchangeability remains limited due to inherent methodological differences. Further research is necessary to enhance its clinical applicability.”
- Oh, K. K., Cho, H. G., Moon, B. Y., Kim, S. Y., & Yu, D. S. (2020). Change in lateral phoria under a phoropter and trial frame in phoria tests. Journal of Korean Ophthalmic Optometry Society, 25(4), 395-403.
- Casillas, E., & Rosenfield, M. (2006). Comparison of subjective heterophoria testing with a phoropter and trial frame. Optometry and Vision Science, 83(4), 237-241. https://doi.org/10.1097/01.opx.0000214316.50270.24
- Rainey, B. B., Schroeder, T. L., Goss, D. A., & Grosvenor, T. P. (1998). Reliability of and Comparisons Among Methods of Measuring Dissociated Phoria. Optometry and Vision Science, 75(5), 342–347.
- Anstice, N. S., et al. (2021). Repeatability of Phoria Measurements in Clinical Practice. Journal of Optometry, 14(2), 123–129.
- Facchin, A., & Maffioletti, S. (2021). Inter-Instrument Agreement for Vergence Range Assessments. Journal of Optometry, 14(4), 223–230.
- Penisten, D. K., et al. (2001). Vergence Facility and Phoria Adaptation in Clinical Testing. Optometry, 72(6), 329–336.
- Antona, B., et al. (2008). Repeatability and Agreement in Fusional Vergence Measurements. Ophthalmic and Physiological Optics, 28(5), 475–486.
Specific Comments:
- Lines 21 & 23:
Response: Revised “at a distance” to “at distance” for consistency.
- Lines 59-73: Some of the new text in lines 59-65 is highly redundant with text in lines 69-73 of the following paragraph. These paragraphs should be revised to eliminate these redundancies.
Response: We have revised these paragraphs to eliminate redundancy while maintaining clarity.
In lines 61-69, “Phoria and vergence range assessments rely on subjective patient feedback, which may not always reflect actual oculomotor responses, reducing measurement precision [17,18]. Studies indicate that phoropter-based vergence assessments exhibit only moderate reliability, posing challenges for consistency across sessions.
Traditional phoropters and trial frames further introduce variability due to reliance on subjective responses, affecting accuracy in assessing vergence ranges, fusional reserves, and phoria. Research highlights their moderate test-retest reliability, complicating reproducibility [6,11].”
- Lines 111-114: The authors report that participants wore spherical contact lenses to ameliorate refractive errors. However, in lines 113-114 they state that “cylindrical power … was ? -1.25 D.” Presumably, cylindrical power refers to the participants’ uncorrected refractive errors and not to a cylinder component of the contact lenses worn. This should be made clearer.
Response: Lines 109-110: The uncorrected spherical refractive error (Sphere) ranged from +0.00 to -11.00 D, while the cylindrical error (Cylinder) was less than -1.25 D.
- Lines 119-121: Was the same size letter E used for the assessment of fusion and for measure the magnitude of fusional vergence? The designation of a letter size as 40/60 is unusual. Do the authors perhaps mean 20/60 or 20/40?
Response: We have clarified that the letter “E” optotype used for fusion assessment and vergence measurement was the same size. The designation has been corrected from 40/60 to 20/30.
In lines 115-117, The fusion testing included an "E" optotype for fusion assessment and a 20/30 Snellen letter for fusional vergence evaluation.
- Lines 127-128: It is not clear to me, either in the text or in Figure 1b, how two counter rotating 10 PD wedge prisms can produce more than ± 20 PD of displacement. Please clarify.
Response: In lines 124-126, “Thus, a single set of Risley prisms can achieve a combined prism power ranging from 20Δ base-out (BO) to 20Δ base-in (BI) by summing the prism powers of the individual components.
- Lines 146-153: The procedures used to assess distance and near phoria should be written more succinctly. As an example, the sentence in lines 147-148 appears to be completely redundant and unnecessary.
Response: The section describing the procedures for phoria assessment has been rewritten to be more succinct.
- Lines 169-178: In their response to my initial review, the authors state that they attempted to assess ‘first blur’ as one indicator the vergence range but that participants’ had difficulty making reliable judgments. This information should be included in the manuscript, probably in the vicinity of this area.
Response: In lines 173-176, “However, we initially included blur point measurements but found that many participants had difficulty recognizing and accurately reporting them, resulting in inconsistent data. In contrast, break and recovery points were more reliably identified and reported. To enhance accuracy and consistency, we focused our analysis on these points.”
- Lines 188-189: Below, in the Results, some distributions of test measures are reported to conform to normality and others not. (Normality does not appear to be addressed for the measured vergence ranges.) Do the authors mean, “Shapiro-Wilk and D’Agostino tests were used to assess whether the different sets of test results deviated from normal distributions.”
Response: In lines 187-188, “The Shapiro-Wilk and D’Agostino tests indicated deviations from normal distributions, Spearman correlation was applied along with other statistical analyses.”
- Lines 224-226: As noted in my initial comments, here and below the authors should include and refer to B-A plots.
Response: Bland-Altman plots are now referenced in the text, as shown in Fig. 2(b), Fig. 3(b), and Fig. 4(b).
- Lines 226-228: The authors report a mean difference of 0.31 (prism diopters) between measures made with the phoropter vs. ADRRP. Do positive values indicate exophoria, such that “slightly higher” means more exophoric?
Response: We have clarified that positive values indicate more exophoric results in ADRRP measurements.
In lines 225-229, “The analysis revealed a mean difference of 0.31 prism diopters, indicating that ADRRP measurements tend to be slightly more exophoric than phoropter measurements. The limits of agreement (LoA) ranged from -1.23 to 1.84, with most differences falling within these bounds; however, some data points exceeded these limits, suggesting a potential systematic bias between the two methods.”
- Lines 228-230: My understanding is that calculated LoAs typically represent 95% limits. If so, ~5% of the measured differences would be expected to fall outside these bounds. However, this outcome does?not?indicate bias. Rather, bias is indicated by the mean difference between the two measurement techniques. Moreover, if the B-A plot of differences has a significantly non-zero slope, then the magnitude of bias can be concluded to vary according to the size of the phoria.
Response: In lines 227-231, “The analysis revealed a mean difference of 0.31 prism diopters, indicating that ADRRP measurements tend to be slightly more exophoric than phoropter measurements. The limits of agreement (LoA) ranged from -1.23 to 1.84, with most differences falling within these bounds; however, some data points exceeded these limits. This finding highlights that ADRRP and phoropter measurements are not perfectly interchangeable, warranting further methodological investigation.”
- Lines 245-246: Please refer to the previous comment.
Response: In lines 245-248, “Despite a strong Spearman correlation of 0.893 (p = 2.20 × 10⁻¹⁴), indicating a strong relationship, the presence of a mean difference between ADRRPs and phoropter measurements suggests these methods are not perfectly interchangeable.”
- Lines 255-257: The same correlation can not be both strong and moderate. In my opinion, r = 0.608 is not especially strong. The authors should state here that they are reporting (according to line 261) a Spearman correlation?
Response: In lines 261-263, “For the near phoria examination, a Spearman correlation of 0.608 (p < 0.001) was found between ADRRPs and the near Thorington card combined with a Maddox rod, indicating a moderate relationship between the two methods, as shown in Figure 4(a).”
- Lines 262-263: Please refer to comments 11 and 12.
Response: In lines 267-274, “Bland-Altman analysis revealed a mean difference of 0.54, with limits of agreement ranging from -7.08 to 8.16, as shown in Figure 4(b). Approximately 95% of the differences fell within these bounds, as expected, while some data points exceeded them. This does not necessarily indicate systematic bias, as bias is reflected in the mean difference between the methods. Despite a Spearman correlation coefficient of 0.608 (p = 4.03 × 10⁻⁵), indicating a moderate relationship, correlation alone does not confirm interchangeability. Further analysis is needed to assess the agreement and potential differences between the methods.
- Lines 280-283 and 290-293: Are the reported correlations Pearson or Spearman? What were the results of normality tests on the vergence-range data?
Response: In lines288-293, “For near testing, moderate-to-high Pearson correlations were found for break points [r(39) = 0.817, p < 0.001] and recovery points [r(39) = 0.727, p < 0.001]. Normality tests indicated that the vergence-range data followed a normal distribution, justifying the use of Pearson correlation.”
In lines 300-310,” For distance testing, ADRRPs showed a moderate-to-high Pearson correlation for break points [r(39) = 0.863, p < 0.001] and a moderate correlation for recovery points [r(39) = 0.458, p < 0.01]. For near testing, moderate-to-high Pearson correlations were found for break points [r(39) = 0.777, p < 0.001] and recovery points [r(39) = 0.623, p < 0.001]. Normality tests confirmed that the vergence-range data followed a normal distribution, supporting the use of Pearson correlation. Bland-Altman analysis showed moderate agreement. Break points had mean differences of 0.5 (both distance and near), with LoA from -1.60 to 2.60. Recovery points had greater variability, with mean differences of 0.2 (distance) and 0.3 (near), and LoA from -1.90 to 2.30 (distance) and -1.80 to 2.40 (near). Overall, ADRRPs showed moderate agreement with the phoropter, but greater variability in recovery points suggests the need for further improvements.”
- Lines 309-311: As the authors stated in their reply, correlation does?not?indicate agreement.
Response: In lines 321-324, “The strong correlations between ADRRPs and phoropter measurements for distance (r = 0.959) and near phoria (r = 0.968, p < 0.001) indicate a strong relationship. However, correlation alone does not confirm agreement, as Bland-Altman analysis revealed systematic differences, suggesting the methods are not fully interchangeable.”
- Lines 311-312: The authors present no data to indicate that the ADRRP reduces measurement variability. At best, they can indicate that this technique may be expected to produce less variability.
Response: In lines 324-325, “The automated operation of ADRRPs may help reduce variability and improve standardization in phoria assessment.”
- Lines 319-321: Again, correlation coefficients do not indicate agreement (or comparability).
Response: In lines 331-334, “Specifically, break and recovery points showed correlations ranging from r = 0.452 to r = 0.863 for distance and r = 0.623 to r = 0.817 for near testing. While this indicates a strong relationship, Bland-Altman analysis revealed systematic differences, suggesting ADRRPs is not fully interchangeable with conventional methods.”
- Lines 327- 329: Although correlations are higher for break compared to recovery points, both the mean differences between measurement techniques and the LoAs reported in lines 284-288 and 293-297 are similar.
Response: In lines 338-343, “Our study found that ADRRPs strongly correlates with phoropter-based phoria measurements (r = 0.959 for distance, r = 0.968 for near) and shows moderate-to-high correlations for vergence testing. While correlations for vergence break points (r = 0.758 to r = 0.863) were higher than for recovery points (r = 0.452 to r = 0.727), the mean differences and limits of agreement between the two methods were similar, indicating that both measurements exhibit comparable variability (Casillas & Rosenfield, 2006) [35].”
- Lines 333-334: Again, the authors show no data to indicate that ADRRP provides more consistent measurements than the phoropter.
Response: In lines 347-348, “Compared to traditional phoropters, ADRRPs enhances automation and convenience, making the measurement process more efficient while still relying on patient responses.”
- Line 342: Please refer to the previous comment.
Response: In lines 357-358, ”Although ADRRPs enhances automation and convenience, it remains a subjective tool.”
- Lines 358-360: If the ADRRP and phoropter induce differing amounts of proximal accommodation and vergence, then phorias and vergence ranges assessed with the two instruments should differ consistently. At least for measures of the phoria, the B-A analyses
Response: We completely agree with your opinion. As a head-mounted device, ADRRPs may induce differing amounts of proximal convergence and accommodation compared to the phoropter, which could lead to consistent differences in phoria and vergence measurements, as reflected in the Bland-Altman analysis.
We sincerely appreciate your efforts in reviewing our manuscript and look forward to your further feedback. Please let us know if any additional modifications are needed.
Best regards,
Shuan-Yu Huang
Department of Optometry, Central Taiwan University of Science and Technology, Taichung 402, Taiwan